# On the Ranking of Human Pose Confidence

## Abstract

While 2D human pose estimation (HPE) has achieved strong advances in keypoint localization, the ranking of pose confidence scores has received little attention. These scores are central to evaluation protocols such as mean Average Precision (mAP) and to applications like pose selection, where predictions are ordered or filtered based on their scores. Yet, the quality of these rankings is often suboptimal, limiting overall performance. In this paper, we ask whether explicitly optimizing the ranking of confidence scores, without altering keypoint coordinates, can improve HPE. To this end, we formulate confidence ranking as a pairwise ordering problem, which, to our knowledge, has not been directly explored in HPE. We further propose a rank loss that upper bounds the negative expected likelihood of correct orderings and guarantees that reducing the loss leads to higher-quality rankings. To validate this formulation, we present Ranked Confidence Net (RCNet), a lightweight module with only 0.07M parameters that refines confidence rankings post hoc while leaving keypoints unchanged. RCNet serves as both a conceptual demonstration of the value of ranking and a practical tool with negligible computational cost. Experiments on COCO show consistent improvements across strong HPE baselines, with an average gain of 0.7 mAP (ranging from 0.3 to 1.8), and consistent gains on CrowdPose. These results establish confidence ranking, independent of coordinate refinement, as an effective and previously overlooked direction for advancing human pose estimation.

## 1 Introduction

2D human pose estimation (HPE) aims to localize human keypoints from images and has seen rapid progress with increasingly accurate localization methods. Beyond localization, confidence scores are equally important. In safety-sensitive applications such as autonomous driving and human–robot collaboration (Abdar et al., 2021; Su et al., 2023; Scherf et al., 2024), downstream decisions require reliable estimates to select trustworthy poses. Evaluation metrics also emphasize confidence: mean Average Precision (mAP), the dominant benchmark in HPE, is confidence-sensitive because it depends on how well predicted scores align with the true quality of poses.

Crucially, it is not only the reliability of individual scores that matters, but their relative ordering. A case study in Fig. 1 shows how misranking can severely harm performance: assigning higher confidence to a false positive (P4) than to true positives (P2 and P3) causes a large drop in AP, even though all keypoint localizations are identical. This reveals that the effectiveness of confidence in HPE depends fundamentally on ranking quality, a factor that has received little systematic study.

Despite their importance, confidence scores and their ranking have been largely overlooked in HPE, or treated only with heuristic refinements that lack theoretical grounding (Geng et al., 2021; Cheng et al., 2023; Gu et al., 2023). For example, Geng et al. (2021) and Cheng et al. (2023) apply model-specific Multi-Layer Perceptrons (MLPs) to adjust confidence values, but these are ad hoc designs that regard confidence as a secondary detail and provide no analysis of its impact on performance. Gu et al. (2023) represents the first formal study of confidence in HPE, emphasizing that ranking is crucial for pose performance. However, its analysis is limited to methods that represent pose predictions as Gaussian distributions (Xu et al., 2022; Li et al., 2021), restricting its applicability to a narrow subset of architectures, namely top-down approaches.

In this paper, we argue that the central role of confidence in HPE lies in producing the correct ranking of predicted poses. To capture this, we cast confidence refinement as a probabilistic ranking problem.

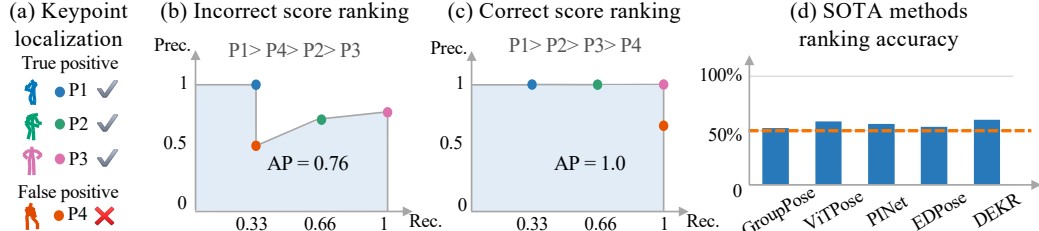

Figure 1: Impact of confidence ranking on AP. (a) The keypoint localization with P1, P2, P3 matched as true positive and P4 matched as false positive. (b) An example of incorrect confidence ranking. Ranking a false positive (P4) before true positives leading to 0.76 AP. (c) Ranking all true positives before the false positive, achieving perfect 1.0 AP. (d) Ranking accuracy of several state-of-the-art HPE methods. The results show that many existing models often fail to rank predictions correctly, with accuracy close to random (orange dashed line). Rec. indicates recall. Prec. indicates precision.

We begin by formulating the task as pairwise ranking, defining the probability that a model correctly orders a pair of poses. The objective is to maximize the expected pairwise ranking accuracy across all pose pairs, which in turn leads to an optimal global ranking. From this formulation, we derive an upper bound on the expected negative log-likelihood of correct orderings (Theorem 1), yielding a closed-form rank loss. Our theoretical analysis further shows that minimizing this loss is bound to improve ranking accuracy in a given rate (Theorem 3) and guarantees convergence (Cor. 6, see Appx. J.3), properties that are desirable in algorithm design. Unlike the strategy of Gu et al. (2023), which depends on Gaussian-based pose representations, our formulation is architecture-agnostic and can be seamlessly applied across different HPE pipelines.

To assess the impact of confidence ranking, we ask whether performance can be improved by refining the rank of scores alone, without altering localization or introducing complex modules. To this end, we design a minimal module that applies a principled ranking loss while keeping keypoint coordinates fixed. Concretely, we introduce Ranked Confidence Net (RCNet), a lightweight plug-and-play module with only 0.07M parameters that refines pose-level confidence scores to maximize pairwise ranking accuracy. Its simplicity ensures that the observed gains come from improved ranking rather than from architectural complexity or additional capacity.

We conduct extensive experiments on the COCO and CrowdPose datasets using a wide range of state-of-the-art HPE baselines spanning top-down, one-stage, and bottom-up paradigms. On COCO, RCNet delivers consistent improvements, with an average gain of 0.7 mAP and up to 1.8 mAP. On CrowdPose, where heavy occlusion makes localization the main bottleneck, RCNet still provides consistent gains when keypoint coordinates are held fixed. These results demonstrate that even a small and simple module can reliably improve performance, indicating that future progress in HPE may come not only from further localization refinements, already close to saturation in many settings, but also from more principled treatment of confidence estimation. This underscores the need for deeper study of confidence as a key direction for advancing HPE.

In summary, our contributions are:

- We present a principled mathematical formulation of pose-confidence ranking as a pairwise ordering problem, offering a new perspective for analyzing and improving confidence in HPE. The formulation is general and architecture-agnostic, enabling a clear connection between ranking quality and widely used evaluation metrics such as mAP.

- We introduce a rank loss that explicitly maximizes the likelihood of correct pose orderings, supported by rigorous theoretical guarantees. The rank loss is both practical and impactful: it directly improves evaluation metrics such as mAP, is architecture-agnostic, and provides improvement guarantees that make it a reliable building block for future confidence modeling.

- We design Ranked Confidence Net (RCNet), a lightweight, attention-based, plug-and-play post-hoc module with only 0.07M trainable parameters and negligible inference cost, which refines confidence rankings and yields consistent gains across diverse HPE baselines.

## 2 RELATED WORKS

**2D Human Pose Estimation** Early top-down pipelines detect a person bounding box and then apply a single-person pose estimator, evolving from multi-stage cascades such as Hourglass (Newell et al., 2016) to high-resolution backbones like HRNet (Sun et al., 2019) and SimpleBaseline (Xiao et al., 2018). Bottom-up methods detect all keypoints jointly before grouping them into skeletons, with approaches including OpenPose, Associative Embedding (Newell et al., 2017), HigherHRNet (Cheng et al., 2020), and DEKR (Geng et al., 2021). One-stage frameworks (Liu et al., 2023b; Yang et al., 2023) integrate detection and grouping in a single pass (e.g., RTMPose (Jiang et al., 2023), YOLO-Pose (Maji et al., 2022)). Recent transformer-based designs such as ViTPose (Xu et al., 2022) and hybrid decoupling networks (Cheng et al., 2023) set new accuracy records, though improvements in mAP have slowed, motivating a closer study of confidence scores.

**Confidence Estimation** Confidence calibration aligns predicted scores with empirical correctness. Classical approaches include Platt scaling (Platt et al., 1999) and temperature scaling (Guo et al., 2017), with extensions such as focal loss (Lin et al., 2017) and Bayesian binning (Naeini et al., 2015). Calibration has been widely studied in classification (Guo et al., 2017; Mukhoti et al., 2020; Ghosh et al., 2022), detection Pathiraja et al. (2023); Qi Fan et al. (2022), and segmentation (Wang et al., 2023; Pau de Jorge et al., 2023), but only recently in HPE. Gu et al. (2023) adapted temperature scaling to Gaussian heatmap heads, showing mAP improvements but with limited applicability to heatmap-based top-down models.

**Learning to Rank** Learning-to-Rank (LTR) methods learn scoring functions for ordering items, including pairwise ranking with SVM-Rank (Joachims, 2002), Bayesian Personalized Ranking (Rendle et al., 2012), RankNet's cross-entropy loss (Burges et al., 2005), lambda-based pseudo-gradients Burges et al. (2006); Burges (2010a), LambdaMART (Burges, 2010b), and listwise relaxations (Qin et al., 2010; Pobrotyn & Białobrzeski, 2021). Our setting differs in that we apply task-specific, post-hoc re-ranking of pose confidences after baselines produce keypoints and initial scores. While some general LTR methods can transfer, they are not directly effective for HPE. Related work in detection ranking (Chen et al., 2020; Qian et al., 2020) provides additional context, with further ablations in Appendix E.

## 3 PRELIMINARY

### 3.1 PROBLEM SETTING

Given an image $\mathbf{I} \in \mathbb{R}^{3 \times h \times w}$ containing multiple persons, 2D HPE seeks to learn a pose estimator $\mathcal{M}$ that predicts $k$ keypoints for each person. Along with keypoint coordinates, $\mathcal{M}$ also assigns a confidence score $s \in [0,1]$ to each pose $\boldsymbol{p} \in \mathbb{R}^{k \times 2}$, where the last dimension stores the $(x, y)$ locations. The estimator therefore outputs a set of pose predictions together with their associated confidences:

$$(\boldsymbol{p}_1, \boldsymbol{p}_2, \ldots, \boldsymbol{p}_N), (s_1, s_2, \ldots, s_N) = \mathcal{M}(\mathbf{I}). \tag{1}$$

Most existing HPE baselines primarily optimize the localization accuracy of $\boldsymbol{p}$, while the confidence scores $s$ are often handled with simple heuristics. Top-down methods fuse the detector score with the average of keypoint heatmap peaks; bottom-up models aggregate selected keypoint confidences after grouping; and one-stage transformer-based approaches often use the decoder's query score. In contrast, this work explicitly focuses on studying and improving $s$ through principled ranking.

### 3.2 EVALUATION METRIC AND GROUND-TRUTH ORDER OF CONFIDENCE SCORES

Evaluation in HPE begins at the keypoint level, where the standard metric is Object Keypoint Similarity (OKS) Lin et al. (2014). OKS provides a continuous measure of keypoint accuracy based on distance, visibility, and keypoint-specific tolerances, as defined in Eq. 2, where $v_i$ indicates visibility, $\sigma_i$ is the predefined variance for keypoint $i$, $p_i$ is the predicted location, $p_i^*$ is the ground-truth, and $c_i$ is a dataset-defined normalization constant:

$$\text{OKS} = \sum_{i=1}^{k} v_i \exp\left(-\frac{\|p_i - p_i^*\|}{2\sigma_i^2 c_i^2}\right). \tag{2}$$

Table 1: The ranking accuracy of existing methods on COCO val set.

| Method | HRNet | ViTPose | CID | GroupPose | EDPose | LogoCap | HighierHRNet | SWAHR | DEKR | PINet |
|--------|-------|---------|-----|-----------|--------|---------|--------------|-------|------|-------|
| Rank Acc. | 59.12% | 58.39% | 58.37% | 52.00% | 53.31% | 57.49% | 57.21% | 63.56% | 59.59% | 56.04% |

By design, OKS values lie in $[0, 1]$, making them natural surrogate ground-truth scores $s^*$. Higher OKS indicates higher relative quality and thus determines the ground-truth ordering among poses. Using these values, HPE is typically evaluated at the person level with two summary metrics: mean Average Precision (mAP) and mean Average Recall (mAR), computed over $T = 10$ OKS thresholds $\{\tau_t\}_{t=1}^T$ evenly spaced between 0.50 and 0.95.

**Mean Average Precision (mAP)**   mAP is defined in Eq. 3, where $\mathcal{I}$ is the indicator function and $i'$ indexes predictions after sorting by confidence $s_i$. This mirrors the area under the Precision–Recall curve (AUPRC) in classification. Crucially, because sorting is performed before integration, mAP is sensitive to the ranking of confidence scores, while their absolute magnitudes play no role (see Fig. 1 for illustration).

$$\text{mAP} = \frac{1}{T} \sum_{t=1}^{T} \sum_{i'=1}^{N} \frac{\mathcal{I}(\text{OKS}_{i'} > \tau_t)}{N} \cdot \frac{\sum_{j=1}^{i'} \mathcal{I}(\text{OKS}_j > \tau_t)}{i'}. \tag{3}$$

**Mean Average Recall (mAR)**   mAR, given in Eq. 4, aggregates recall across thresholds without sorting, making it insensitive to confidence ordering. This parallels the area under the ROC curve (AUROC).

$$\text{mAR} = \frac{1}{T} \sum_{t=1}^{T} \sum_{i=1}^{N} \frac{\mathcal{I}(\text{OKS}_i > \tau_t)}{N}. \tag{4}$$

Between the two, mAP is the primary benchmark, as it evaluates both coverage and ranking quality, rewarding methods that place correct poses at the top of the list. In contrast, mAR merely answers whether correct poses were detected, regardless of their position. For this reason, most works emphasize mAP as the headline metric and report mAR only as a secondary measure.

### 3.3 Current Status of Confidence Ranking

Although confidence ranking is crucial for HPE, most existing methods do not optimize it explicitly. As shown in Tab. 1, the pairwise ranking accuracy of predictions matched to ground-truth poses hovers around 60% across common baselines, with the strongest (SWAHR) only slightly higher. For comparison, random ranking yields 50% accuracy. This persistent gap highlights the limitations of current approaches and motivates our method.

## 4 Proposed Method

Existing HPE models produce confidence scores, but because these scores are not explicitly trained to reflect the correct ordering of pose quality, the resulting ranking accuracy is only slightly better than random when evaluated pairwise. To address this, we design a lightweight module that reorders the pose predictions of a pretrained estimator $\mathcal{M}$ by refining their initial confidence scores $s$. Since current ranking accuracy is nearly random, we first analyze the failure modes of random ranking, which motivates our formulation.

**Random Ranking**   The worst case arises when confidence scores are assigned randomly, i.e., $s$ is sampled i.i.d and independent to poses. For a random sampled pair of poses $(i, j)$, the probability of correct ordering is

$$\mathbf{P}_{\text{correct}}^{ij} = \mathbf{P}(s_i > s_j, s_i^* > s_j^*) + \mathbf{P}(s_i < s_j, s_i^* < s_j^*). \tag{5}$$

Assuming a fixed, well-trained model $\mathcal{M}$, the ground-truth scores $s^*$ can be regarded as i.i.d. Expanding with Bayes' rule then gives

$$\mathbf{P}_{\text{correct}}^{ij} = \mathbf{P}(s_i > s_j \mid s_i^* > s_j^*) \mathbf{P}(s_i^* > s_j^*) + \mathbf{P}(s_i < s_j \mid s_i^* < s_j^*) \mathbf{P}(s_i^* < s_j^*). \tag{6}$$

With i.i.d. sampling, the two conditional terms are equal and reduce to $\frac{1}{2}$, yielding $\mathbf{P}^{ij}_{\text{correct}} = 0.5$, the same as a coin flip.

This analysis shows that the i.i.d. assumption of $s'_i$ and $s'_j$ locks ranking accuracy at chance level. To overcome this, we propose a two-step approach: (1) generate a refined score $s'$ from a parametric distribution $p_\theta(s'; s, \boldsymbol{p})$ so that $s'_i$ and $s'_j$ are no longer identically distributed, and (2) learn parameters $\theta^*$ that maximize $\mathbf{P}^{ij}_{\text{correct}}$ induced by $s'$. This leads naturally to our rank loss.

## 4.1 RANK LOSS

**Problem Formulation** We denote the relation between two elements $x$ and $y$ as $x \prec y$, where $\prec$ indicates either $>$ or $<$ and ties are excluded. The training objective in Eq. 7 is the expected negative log-likelihood of predicting the correct pairwise relation. We therefore seek the parameters $\theta^* = \arg\max_\theta(-L(\theta))$, which maximize this likelihood or, equivalently, minimize the loss $L(\theta)$:

$$
\begin{aligned}
L(\theta) &= -\mathbb{E}_{ij}[\log \mathbf{P}^{ij}_{\text{correct}}] = -\mathbb{E}_{ij}[\log \mathbf{P}_\theta(s'_i \prec s'_j, s^*_i \prec s^*_j)], \\
&= -\mathbb{E}_{ij}[\log \mathbf{P}_\theta(s'_i \prec s'_j | s^*_i, s^*_j) p(s^*_i) p(s^*_j)], \\
&= -\mathbb{E}_{ij}[\log \mathbf{P}_\theta(s'_i \prec s'_j | s^*_i, s^*_j)].
\end{aligned}
\tag{7}
$$

**Laplacian Assumptions** To obtain a closed-form expression for the loss, we adopt two assumptions:

- The refined confidence $s'$ follows a Laplace distribution, $s' \sim \text{Laplace}(\mu, b)$.
- The distribution mean $\mu$ is Laplace-distributed around $s^*$, i.e. $\mu \sim \text{Laplace}(s^*, b')$.

For the first assumption, the choice of a Laplace distribution is motivated not only by tractability but also by its connection to strictly proper scoring rules in probabilistic forecasting (see App. G). For the second assumption, after discarding poses with very low initial scores $s$, a well-trained HPE baseline should produce scores near their true values $s^*$. A refined score $s'$ should also lie close to $s^*$, focusing the task on ranking relatively high-quality poses.

We further replace $\mu$ with a learnable network $\mu_\theta(s, \boldsymbol{p})$. After training, the refined score $s'$ is estimated by $\mu_\theta(s, \boldsymbol{p})$. Definition 1 introduces the rank loss $l(\theta)$, where $\delta^{ij}_\theta = \mu^i_\theta - \mu^j_\theta$ is the refined score difference, and the score margin $m_{ij} = |\delta^{ij}_\theta| \le c := 1$ since $s' \in [0,1]$

**Definition 1** (Rank Loss). *The rank loss under the Laplacian assumption is defined as*

$$
l(\theta) = \frac{2}{b'}\mathbb{E}_i[|s^*_i - \mu^i_\theta|] + \mathbb{E}_{ij}[g(\delta^{ij}_\theta)],
\tag{8}
$$

*where $g$ is defined below*

$$
g(\delta^{ij}_\theta) = \begin{cases}
\frac{1}{2}(\delta^{ij}_\theta - c)^2 + \frac{1}{2}(\delta^{ij}_\theta)^2 & \text{if } s^*_i > s^*_j, \ \delta^{ij}_\theta \in [0,c] \\
\frac{1}{2}(\delta^{ij}_\theta + c)^2 - \frac{1}{2}(\delta^{ij}_\theta)^2 & \text{if } s^*_i < s^*_j, \ \delta^{ij}_\theta \in [0,c] \\
\frac{1}{2}(\delta^{ij}_\theta - c)^2 - \frac{1}{2}(\delta^{ij}_\theta)^2 & \text{if } s^*_i > s^*_j, \ \delta^{ij}_\theta \in [-c,0) \\
\frac{1}{2}(\delta^{ij}_\theta + c)^2 + \frac{1}{2}(\delta^{ij}_\theta)^2 & \text{if } s^*_i < s^*_j, \ \delta^{ij}_\theta \in [-c,0)
\end{cases}.
\tag{9}
$$

**Properties of Rank Loss** Theorem 1 shows that $l(\theta)$ upper bounds $L(\theta)$. Minimizing $l(\theta)$ thus indirectly optimizes $L(\theta)$, analogous to variational methods that optimize bounds on intractable objectives. Proofs appear in Appx. J.1.

**Theorem 1** (Upper Bound). $L(\theta) \le l(\theta)$ *under the Laplacian assumption.*

The first term of $l(\theta)$ pulls refined scores $s'$ toward their references $s^*$. The second term, $g(\delta^{ij}_\theta)$, enforces correct ordering. When a misranking occurs, $g$ drives $\delta^{ij}_\theta$ toward zero, encouraging a sign flip that corrects the order. When the predicted order is correct, $g$ stabilizes the margin $m_{ij}$ near $c/2$, preventing collapse of refined scores to 0 or 1. Proposition 2 formalizes this, showing that $c/2$ is the expected optimal margin in the uniform case (proof in Appx. J.2)

Figure 2: RCNet with rank loss. RCNet corrects initially misordered pairs (e.g., $\boldsymbol{p}_2$ and $\boldsymbol{p}_4$). (1) Early in training, the loss focuses on flipping the sign of the score difference to fix the order. (2) Once the order is correct, the loss regularizes the margin toward $\frac{c}{2}$. Pairs that are already correct follow step (2) directly.

**Proposition 2** (Specificity of the Margin). *$\frac{c}{2}$ is the expected optimal margin in pairwise ranking under a uniform prior on confidence*

***Remark****.* In practice, $\frac{c}{2}$ serves as a safe default margin when no prior information is available. For known baselines or special setups, the margin can be tuned as a hyperparameter. An ablation study is provided in Appx. D

Finally, Theorem 3 guarantees that reducing $l(\theta)$ always improves pairwise ranking accuracy, ensuring convergence of $\mathbf{P}_{\text{correct}}$.

**Theorem 3** (Improvement of Ranking Accuracy). *If $l(\theta)$ decreases at iteration $t$, then $\mathbf{P}_{correct} = \mathbb{E}_{ij}[\mathbf{P}_{correct}^{ij}]$ increases at a rate of at least:*

$$o(\min\{\mathbb{E}_i[|s_i^* - \mu_t^i|], \ \mathbb{E}_{ij}[(\delta_t^{ij})^2 - (\delta_{t-1}^{ij})^2]\}),$$

*where $\mu_t^i$ and $\delta_t^{ij}$ denote values at iteration $t$.*

The theorem shows that every reduction in the loss guarantees an increase in ranking accuracy, with the rate of improvement lower-bounded by a given term. Here, $\mu_t$ denotes $\mu_\theta^i$ and $\delta_t^{ij}$ denotes $\delta_\theta^{ij}$ at iteration $t$. The rate also highlights that ranking accuracy depends on two terms. The first term aligns the predicted scores with the surrogate ground-truth $s^*$ that induces the target ordering. Reducing this estimation error increases ranking accuracy. The second the term is the variance of the refined score gaps. If this variance becomes too small, the model cannot confidently separate poses, leading to reduced pairwise accuracy. As a direct corollary, the expected ranking accuracy $\mathbf{P}_{\text{correct}}$ is guaranteed to converge whenever the loss is decreasing or convergence, since the $\mathbf{P}_{\text{correct}}$ is bounded by 1 and increases in a given rate.

### 4.2 RCNET

With the theoretical foundations established, we now instantiate the neural function $\mu_\theta(s, \boldsymbol{p})$ for pairwise confidence refinement. Unlike per-pose correction, the model must capture dependencies across all detected poses. A self-attention block is a natural choice: it treats the set of poses as a sequence, models their interactions, and satisfies the gradient-Lipschitz continuity condition, which is a standard assumption that ensures a stable decrease of the loss and underpins convergence guarantee.

We therefore introduce Ranked Confidence Net (RCNet), a lightweight module consisting of a one-head self-attention block with two additional linear layers (head and tail). The full pipeline is shown in Fig. 2.

Formally, RCNet first extracts individual features from the sequence of scores $[\mathbf{s}] = [s_1, \ldots, s_n]$ and poses $[\boldsymbol{p}] = [\boldsymbol{p}_1, \ldots, \boldsymbol{p}_n]$ through a feature extractor $\phi$

$$\mathbf{e} = \phi([\mathbf{s}], [\boldsymbol{p}]). \tag{10}$$

These features are then processed by $\mathbf{a} = \mathcal{A}(\mathbf{e})$, where $\mathcal{A}$ is a self-attention encoder. Finally, a lightweight MLP head with a sigmoid activation outputs the refined scores: $\mathbf{s}' = \text{sigmoid}(\mathcal{H}(\mathbf{a}))$. In compact form, RCNet is expressed as:

$$\mathcal{R}_\theta([\mathbf{s}], [\boldsymbol{p}]) = \text{sigmoid} \circ \mathcal{H} \circ \mathcal{A} \circ \phi([\mathbf{s}], [\boldsymbol{p}]). \tag{11}$$

Table 2: Evaluation on COCO val set. "+Ours" indicates performance after applying our method. Values in blue show the improved performance introduced by our method.

| Group | Method | Backbone | #Params(M) | mAP | $AP^{50}$ | $AP^{75}$ | $AP^M$ | $AP^L$ | mAR |
|---|---|---|---|---|---|---|---|---|---|
| Top-down | HRNet (Sun et al., 2019) | HRNet-w32 | 28.59 | 74.4 | 90.5 | 81.9 | 70.8 | 81.0 | 79.8 |
| | +Ours | – | 0.07 | 74.8 | 90.6 | 82.3 | 71.3 | 81.1 | 79.8 |
| | ViTPose (Xu et al., 2022) | ViT-base | 89.99 | 75.8 | 90.7 | 83.2 | 72.3 | 82.6 | 81.1 |
| | +Ours | – | 0.07 | 76.1 | 90.7 | 83.5 | 72.6 | 82.8 | 81.1 |
| One-stage | CID (Wang & Zhang, 2022) | HRNet-w32 | 29.42 | 69.9 | 88.5 | 76.5 | 64.0 | 78.9 | 75.3 |
| | +Ours | – | 0.07 | 70.3 | 88.6 | 76.7 | 64.4 | 79.7 | 75.3 |
| | GroupPose (Liu et al., 2023a) | ResNet-50 | 50.22 | 72.1 | 89.6 | 79.1 | 67.0 | 79.8 | 80.0 |
| | +Ours | – | 0.07 | 72.4 | 89.6 | 79.3 | 67.4 | 80.1 | 80.0 |
| | EDPose (Yang et al., 2023) | ResNet-50 | 50.57 | 71.6 | 89.8 | 78.6 | 66.2 | 79.7 | 79.3 |
| | +Ours | – | 0.07 | 72.0 | 89.9 | 78.7 | 66.7 | 80.0 | 79.3 |
| Bottom-up | LogoCap(Xue et al., 2022) | HRNet-w32 | 36.05 | 69.6 | 87.5 | 75.9 | 64.0 | 78.0 | 73.6 |
| | +Ours | – | 0.07 | 70.1 | 87.7 | 76.3 | 64.5 | 78.9 | 73.6 |
| | HigherHRNet (Cheng et al., 2020) | HRNet-w32 | 28.70 | 67.4 | 86.6 | 73.3 | 61.7 | 76.3 | 71.8 |
| | +Ours | – | 0.07 | 68.1 | 86.6 | 73.8 | 62.1 | 77.4 | 71.8 |
| | SWAHR(Luo et al., 2021b) | HRNet-w32 | 28.70 | 68.9 | 88.0 | 74.9 | 63.0 | 77.4 | 73.7 |
| | +Ours | – | 0.07 | 69.5 | 88.0 | 75.3 | 63.5 | 78.7 | 73.7 |
| | DEKR(Geng et al., 2021) | HRNet-w32 | 29.60 | 67.1 | 87.9 | 74.1 | 61.5 | 76.1 | 72.9 |
| | +Ours | – | 0.07 | 69.0 | 86.7 | 74.8 | 62.7 | 78.6 | 72.9 |
| | PINet(Wang et al., 2021) | HRNet-w32 | 31.30 | 67.4 | 86.8 | 74.0 | 62.5 | 76.3 | 73.4 |
| | +Ours | – | 0.07 | 69.2 | 87.1 | 75.2 | 63.5 | 78.4 | 73.4 |

# 5 EXPERIMENTAL RESULTS

**Implementation Details** To satisfy the GLC condition, both $\mathcal{H}$ and $\phi$ are implemented as single linear layers. All weights are initialized with Kaiming uniform He et al. (2015), consistent with the second assumption. For the third assumption, training uses a 250-step linear warm-up that scales the learning rate from 0 to $1 \times 10^{-4}$, followed by a constant learning rate of $1 \times 10^{-4}$ for the remaining 50 epochs. The parameter $b'$ is fixed at 2.0. Additional details are provided in Appx. B.

**Datasets and Metrics** Experiments are conducted on COCO Lin et al. (2014) and CrowdPose Li et al. (2019). COCO contains nearly 300K annotated training poses, providing a larger dataset, while CrowdPose includes only 85K poses. The primary evaluation metric is the rank-sensitive mAP (see Sec. 3.2). We also report two calibration indicators, Pearson correlation and AUSE-OKS, to assess how well the refined scores align with ground-truth confidence (see Appx. H).

**Baselines** We evaluate RCNet on a wide range of HPE baselines, including top-down (Sun et al., 2019; Xu et al., 2022), bottom-up (Xue et al., 2022; Cheng et al., 2020; Luo et al., 2021a; Geng et al., 2021; Wang et al., 2021), and one-stage (Wang & Zhang, 2022; Liu et al., 2023b; Yang et al., 2023) methods, providing comprehensive coverage. CCNet (Gu et al., 2023) is omitted because no official code is available, and the reported baseline numbers in the paper are inconsistent with official implementations; reproducing them would be misleading.

## 5.1 QUANTITATIVE RESULTS

**COCO Val** Tab. 2 reports the effect of RCNet on the COCO val set. With only 0.07M extra parameters added to each baseline, RCNet improves mAP for every method tested. Top-down and one-stage methods such as HRNet-w32 and CID show steady gains of +0.3–0.4 mAP, while bottom-up systems benefit the most: DEKR and PINet improve by about +1.8 mAP, corresponding to a 2–3% relative jump. These results demonstrate the module's efficiency: sub-percentage parameter growth yields percent-level accuracy gains.

Gains are concentrated where confidence ordering matters most. Columns $AP^{75}$ and $AP^{M/L}$ improve across nearly all methods (e.g., HigherHRNet gains +0.5 $AP^{75}$), showing that RCNet strengthens high-quality pose detections. In contrast, $AP^{50}$ remains nearly unchanged, as the module focuses on refining ranking rather than altering keypoint localization. Recall metrics are also unaffected, confirming that RCNet neither introduces spurious detections nor removes valid ones. Overall, Tab. 2 highlights how a lightweight self-attention head can consistently refine confidence ranking across paradigms. Fig. 6 illustrates two examples, showing how RCNet corrects mis-ordered scores so the

Table 3: Evaluation on CrowdPose test set. "+Ours" indicates performance after applying our method. Values in blue show the improved performance introduced by our methods.

| Group | Method | Backbone | #Params(M) | mAP | $AP^{50}$ | $AP^{75}$ | $AP^E$ | $AP^M$ | $AP^H$ |
|---|---|---|---|---|---|---|---|---|---|
| One-stage | CID (Wang & Zhang, 2022) | HRNet-w32 | 29.42 | 71.2 | 89.8 | 76.7 | 77.9 | 71.9 | 63.8 |
| | +Ours | – | 0.07 | 71.3 | 89.8 | 76.7 | 77.9 | 71.9 | 63.8 |
| | GroupPose (Liu et al., 2023a) | Swin-L | 220.94 | 74.1 | 91.3 | 80.4 | 80.8 | 74.7 | 66.3 |
| | +Ours | – | 0.07 | 74.3 | 91.3 | 80.5 | 81.0 | 74.8 | 66.6 |
| | EDPose (Yang et al., 2023) | ResNet-50 | 50.57 | 69.8 | 88.8 | 75.8 | 76.7 | 70.5 | 61.1 |
| | +Ours | – | 0.07 | 70.0 | 88.8 | 76.0 | 76.9 | 70.7 | 61.4 |
| Bottom-up | SWAHR (Luo et al., 2021b) | HRNet-w48 | 63.91 | 71.6 | 88.5 | 77.6 | 78.9 | 72.4 | 63.0 |
| | +Ours | – | 0.07 | 71.8 | 88.5 | 77.7 | 79.1 | 72.5 | 63.1 |
| | DEKR (Geng et al., 2021) | HRNet-w32 | 29.60 | 64.7 | 85.0 | 69.5 | 72.0 | 65.4 | 56.4 |
| | +Ours | – | 0.07 | 66.0 | 85.7 | 70.6 | 73.5 | 66.7 | 57.9 |
| | PINet (Wang et al., 2021) | HRNet-w32 | 31.75 | 68.9 | 88.7 | 74.7 | 75.4 | 69.6 | 61.5 |
| | +Ours | – | 0.07 | 69.7 | 89.0 | 75.1 | 76.5 | 70.5 | 62.1 |

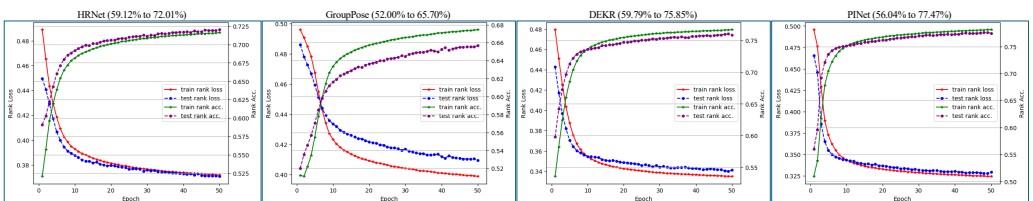

Figure 3: Evolution of ranking loss (left y-axis) and ranking accuracy (right y-axis) across training epochs for each HPE baseline. Methods and test-accuracy improvement are shown above each image. Curves: training loss (red), training accuracy (green), test loss (blue), and test accuracy (purple). Full results are provided in Appx. C.

final ranking aligns with ground truth across back-lit and indoor scenes, yielding more reliable pose selection for downstream use.

**CrowdPose Test** Tab. 3 reports results on CrowdPose test for one-stage and bottom-up baselines. One-stage models gain around +0.2 mAP, while bottom-up methods see larger improvements: DEKR rises by +1.3 to 66.0 mAP, and PINet by +0.8 to 69.7.

Compared to COCO, the improvements are smaller but still consistent. This reflects the method's scope: RCNet operates solely on confidence ordering, leaving localization unchanged. The size of the mAP gain therefore depends on how much of the dataset's error budget comes from mis-ranking rather than mis-localization. On CrowdPose, heavy occlusion makes localization the dominant challenge, which our module does not address. Even so, confidence refinement remains valuable, as it targets a complementary error source and can combine with future localization advances. This underscores the distinct and complementary role of ranking in HPE.

**Ranking Accuracy and Loss** Fig. 3 shows that rank loss decreases rapidly within the first five epochs across all baselines and flattens well before 50 epochs, with training and test curves closely aligned. This stable behavior provides empirical evidence for convergence guarantee. In parallel, ranking accuracy rises from around 50% to 65–75%. For example, HRNet reaches 72.01% (+13 percentage points), GroupPose improves by +13.7 p.p., and DEKR and PINet increase by 16–20 p.p. The accuracy trajectories closely mirror the loss curves: every reduction in loss corresponds to an accuracy gain, validating Theorem 3. The tight match between training and test accuracy further indicates minimal overfitting. Additional results and discussion are provided in Appx. C.

## 5.2 ABLATION STUDIES

**Other Ranking Methods** We compare our approach with representative ranking baselines from learning-to-rank (LTR), information retrieval (IR), and object detection, as shown in the 3rd to 5th columns of Table 4. Due to space limits, we report three methods: RankNet Burges (2010a), NeuralNDCG Pobrotyn & Białobrzeski (2021), and AP Loss Chen et al. (2020); a broader comparison is provided in Appx. E. Results show that only our method achieves consistent gains across diverse

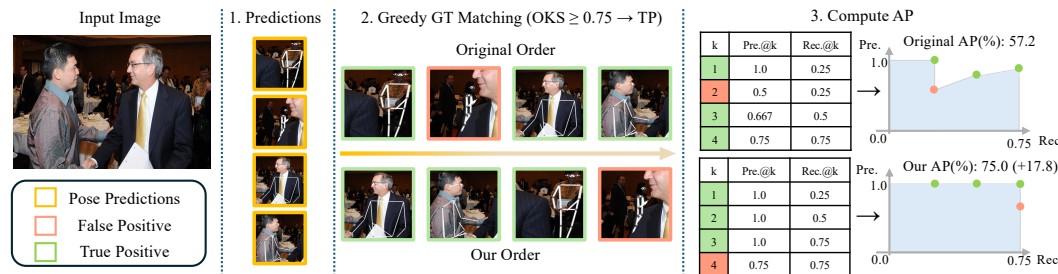

Figure 4: A representative qualitative example of ranking correction on COCO val. From left to right: input image; predictions from the HPE model; COCO's greedy matching after sorted by confidence (with false positives in pink and true positives in green); and the revised ordering showing improved AP for this image.

Table 4: Ablation study of other ranking methods and assumptions over COCO val set. mAP is reported. Best is **bolded**.

| Method | Base Model | Ranking Methods | | | Other Assumptions | | Ours |
|---|---|---|---|---|---|---|---|
| | | RankNet | NeuralNDCG | AP Loss | CRPS | Gaussian | |
| HRNet Sun et al. (2019) | 74.4 | 73.4 | 74.4 | 74.6 | 69.5 | 74.6 | **74.8** |
| ViTPose Xu et al. (2022) | 75.8 | 74.5 | 75.6 | 74.7 | 69.8 | 75.9 | **76.2** |
| GroupPose Liu et al. (2023b) | 72.1 | 71.7 | 71.0 | 61.2 | 62.5 | 72.3 | **72.4** |
| EDPose Yang et al. (2023) | 71.6 | 70.6 | 70.6 | 70.4 | 61.9 | 71.9 | **72.0** |
| DEKR Geng et al. (2021) | 67.1 | 67.6 | 69.0 | 67.7 | 67.9 | 68.6 | **69.0** |
| PINet Wang et al. (2021) | 67.4 | 68.2 | 69.1 | 68.2 | 68.3 | 68.9 | **69.2** |

HPE baselines, whereas the others are less effective. RankNet optimizes local pairwise preferences with a general formulation not tailored to HPE. NeuralNDCG focuses on top-ranked items, while HPE requires accurate pairwise ordering. AP Loss assumes a fixed list length, but candidate poses per image are variable and, in principle, unbounded.

**Laplacian or Gaussian?** We also test variations of the rank loss with probabilistic forecast or alternative distributional assumption in the 6th and 7th columns of Table 4. CRPS, the Continuous Ranked Probability Score, is a strictly proper scoring rule widely used in probabilistic forecasting and motivates our Laplacian choice. However, it underperforms because it is not inherently rank-aware. Replacing the Laplacian with a Gaussian distribution yields slightly worse results, as it removes the implicit margin regularization provided by the Laplacian. Full mathematical details of CRPS and Gaussian appear in Appx. G.

**Training and Inference Cost** All experiments were run on a single RTX 3090 with a batch size of 128 poses. The overhead introduced by RCNet is minimal: training requires only 4.32 seconds per epoch with 665 MB of GPU memory, while inference adds just $2.25 \times 10^{-4}$ ms per pose with 408 MB of memory. These results indicate virtually no additional latency, making the method suitable for real-time deployment.

## 6 CONCLUSION

We introduced a new perspective on 2D human pose estimation by showing that confidence scores matter primarily through their ranking. To capture this, we formulated confidence refinement as a pairwise ranking problem and derived a principled rank loss with strong theoretical guarantees, ensuring that minimizing the loss improves ranking accuracy. To make the approach practical, we proposed RCNet, a lightweight post-hoc module with only 0.07M parameters that integrates seamlessly into diverse HPE baselines. Experiments on COCO and CrowdPose demonstrate consistent improvements, highlighting that confidence ordering, rather than further localization refinements, is now a key bottleneck in HPE. This work shifts attention from saturated localization accuracy to principled confidence ranking, opening a new research direction for pose estimation. Future work

can explore richer distributions beyond Laplacian assumptions and extend ranking-based refinement to video and multimodal scenarios.

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

## A   THE USE OF LARGE LANGUAGE MODELS (LLMs)

We used ChatGPT (GPT-5 Thinking, OpenAI) solely for language editing of the manuscript, including grammar, phrasing, and formatting of author-written text (including LaTeX). The LLM did not generate ideas, methods, experiments, results, or citations; all technical content and claims were authored and verified by us.

## B   ADDITIONAL TECHNICAL DETAILS

Additional implementation details are as follows. The attention module $\mathcal{A}$ is a single-layer Transformer encoder with one head, while the output head $\mathcal{H}$ consists of a single unit. The embedding function $\phi$ is a fully connected layer that projects the pre-processed pose features and initial scores into a 16-dimensional hidden space. We adopt the deterministic, rule-based pre-processing of DEKR Geng et al. (2021). As this step is deterministic, it does not violate the GLC assumption. All experiments are run in PyTorch 2.0.1 on a single RTX 3090 GPU (CUDA 11.9). Results are consistent across runs, so error bars are omitted.

## C   RANKING ACCURACY IMPROVEMENT AND ITS RELATION TO THE AP IMPROVEMENT

We show the full results of ranking accuracy in the following table. RCNet helps all the HPE baselines improve the ranking accuracy. therefore achieving stronger HPE performance.

Table 5: The ranking accuracy of existing methods on COCO val set.

| Method | HRNet | ViTPose | CID | GroupPose | EDPose | LogoCap | HighierHRNet | SWAHR | DEKR | PINet |
|---|---|---|---|---|---|---|---|---|---|---|
| Rank Acc. | 59.12% | 58.39% | 58.37% | 52.00% | 53.31% | 57.49% | 57.21% | 63.56% | 59.59% | 56.04% |
| +Ours | 72.01% | 70.27% | 74.56% | 65.70% | 65.21% | 76.83% | 75.04% | 75.97% | 75.85% | 77.47% |

**Ranking Accuracy and its Impact on mAP** Notably, while ranking accuracy improves consistently across HPE methods, the corresponding mAP gains are smaller, particularly when comparing top-down and bottom-up models. This is largely because the mAP improvement is not affected merely by the ranking accuracy. It also affected by the number of false positives (FPs) ranked above true positives (TPs), which are typically fewer in top-down pipelines as shown in Tab. 6 (the experiment is conducted on COCO val 2017). Therefore, correcting the even same percentage of wrong ranking, the scale of the mAP gain could be different.

For example, we assume a toy dataset contains a single TP. In that situation, AP collapses to a single precision value= TP / (TP+FP). If only 1 FP above TP (common in top-down), correcting this pair raises precision from 0.50 to 1.00, yielding +0.5 AP. In contrast, if there are 4 FP above 1 TP (common in bottom-up), correcting the TP-and-top-FP pair raises precision from 0.20 to 1.00, yielding +0.8 AP.

As shown in the following table, top-down models like HRNet and ViTPose have  13.9% FPs per TP, while bottom-up models like PINet reach  21.8%. Even a modest 2%-10% rise in the FP-to-TP ratio is critical: every extra FP sharply decrease the precision ($1.00 \rightarrow 0.50 \rightarrow 0.33 \rightarrow 0.25 \ldots$). From the table, we clearly see there are around 100 - 700 such FPs. This explains why, despite similar ranking accuracy improvements, top-down models show smaller AP gains—a limitation of the baseline's error distribution, not of our method.

## D   ABLATION STUDY ON OPTIMAL MARGIN

We provide additional ablation on the margin term. While some backbones may slightly benefit from alternative margins (e.g., 0.3 or 0.7), we adopt $m = 0.5$ as a theoretically sound and generalizable default that avoids test-time overfitting. Meanwhile, in the proof of Theorem 3, we show that any margin $m \in (0, c]$ that minimizing the loss, improves ranking accuracy. Empirically, this is validated

Table 6: A statistics of FPs and TPs over different top-down and bottom-up baselines.

| Group | Method | # of Wrongly Ranked FP | Wrong FP per TP |
|---|---|---|---|
| Top-Down | HRNet | 1444 | 0.139 |
| Top-Down | ViTPose | 1333 | 0.159 |
| Bottom-Up | DEKR | 1565 | 0.159 |
| Bottom-Up | PINet | 2165 | 0.218 |

Table 7: Ablation studies over different marginal in rank loss. The best is **bolded**.

| Method | Margin=0.0 | Margin=0.3 | Margin=0.7 | Margin=1.0 | Margin=0.5 (Ours) |
|---|---|---|---|---|---|
| HRNet Sun et al. (2019) | 61.3 | 74.2 | 74.7 | 74.4 | **74.8** |
| ViTPose Xu et al. (2022) | 69.5 | 76.1 | **76.2** | 75.8 | 76.1 |
| GroupPose Liu et al. (2023b) | 21.3 | 71.8 | 71.9 | 71.8 | **72.4** |
| EDPose Yang et al. (2023) | 68.5 | 71.5 | 71.6 | 71.1 | **72.0** |
| DEKR Geng et al. (2021) | 60.0 | **69.0** | **69.0** | 68.9 | **69.0** |
| PINet Wang et al. (2021) | 64.3 | 68.9 | 69.1 | 69.0 | **69.2** |

by our experiments: when $m = 0$, training collapses as the model pushes all scores toward equality; when $m = 0.5$, we consistently achieve the best mAP across all HPE backbones. This supports Theorem 2, which identifies $m = 0.5$ as the optimal choice (or a safe global default) when the explicit HPE baseline is unknown during the real application. However, when applied on specific HPE baseline, the margin could benefit from a further tuning. The Theorem 2 did not deny the existence of such conditional optimality.

# E    ABLATION STUDY ON DIFFERENT RANKING METHODS

In Tab. 8 we introduce several ranking baselines. The table includes traditional LTR methods, such as RankNet Burges et al. (2005) and LambdaRank Burges et al. (2006), Information Retrieval (IR) methods Pobrotyn & Białobrzeski (2021); Qin et al. (2010) and similar ranking method in object detection Qian et al. (2020); Chen et al. (2020).

RankNet is a general-purpose pair-wise method: it models the probability that $s_i' \geq s_j'$ as

$$\mathbf{P}_{ij}(s_i' > s_j') = \frac{1}{1 + e^{-\sigma(s_i' - s_j')}},$$

where $\sigma$ is a temperature hyper-parameter (typically 1). Using the ground-truth ordering between $s_i^*$ and $s_j^*$, its Binary Cross-Entropy objective becomes

$$l_{rn}(\theta) = \mathbb{E}_{ij}[-y_{ij} \log \mathbf{P}_{ij} + (1 - y_{ij}) \log(1 - \mathbf{P}_{ij})],$$

with $y_{ij} = 1$ when $s_i^* \geq s_j^*$ and 0 otherwise.

Although the loss is written in probabilistic form, RankNet treats $s'$ as a deterministic network output. Our method instead assumes $s'$ follows a Laplacian distribution conditioned on the pre-existing score $s$, an assumption well aligned with a post-processing with HPE task that already provide an initial score. This leads to a different optimization landscape: RankNet brings meaningful gains only for DEKR, PINet, and HRNet, whereas our approach improves every baseline consistently.

LambdaRank is metric-specific. It scales each RankNet gradient by the change in Normalized Discounted Cumulative Gain (NDCG) caused by swapping two items, directly steering optimization toward NDCG. Because NDCG rewards concentrates on rank the most significant item at top, while HPE evaluation relies on mAP, which values the correct ordering of all detections, LambdaRank offers limited or even negative benefit for the tested HPE baselines. Several theoretical works show that pairwise models, such as ours, achieve better global ordering accuracy than top-heavy objectives like NDCG, since pairwise comparisons correspond directly to Kendall's tau distance Wauthier et al. (2013); Shah & Wainwright (2018). Consistent with this, information-retrieval methods that implement differentiable relaxations of NDCG (e.g., ApproxNDCG, NeuralNDCG) tend to emphasize performance at the very top of the ranking. Therefore, in our HPE setting they provide only limited gains and generally underperform pairwise losses.

Table 8: Ablation study on different rank loss on COCO val set. mAP is reported. Best is **bolded**.

| Method | Ori. | RankNet | LambdaRank | ApproxNDCG | NeuralNDCG | AP Loss | DR Loss | Ours |
|---|---|---|---|---|---|---|---|---|
| HRNet Sun et al. (2019) | 74.4 | 74.6 | 73.4 | 72.4 | 74.4 | 74.6 | 74.2 | **74.8** |
| ViTPose Xu et al. (2022) | 75.8 | 73.3 | 74.5 | 74.4 | 75.6 | 74.7 | 69.2 | **76.2** |
| GroupPose Liu et al. (2023b) | 72.1 | 68.2 | 71.7 | 70.0 | 71.0 | 61.2 | 62.3 | **72.4** |
| EDPose Yang et al. (2023) | 71.6 | 71.2 | 70.6 | 68.9 | 70.6 | 70.4 | 70.6 | **72.0** |
| DEKR Geng et al. (2021) | 67.1 | 68.6 | 67.6 | 66.9 | 69.0 | 67.7 | 65.1 | **69.0** |
| PINet Wang et al. (2021) | 67.4 | 69.1 | 68.2 | 67.7 | 69.1 | 68.2 | 67.7 | **69.2** |

Table 9: A case study on adding visual features in the RCNet with HRNet-w32 as the vision backbone. Offline indicates the visual features used during training of RCNet are extracted from a frozen vision backbone, which will not be updated during the training. Online indicates the vision backbone updated with the RCNet by both the rank loss and the heatmap loss. The best is **bolded**.

| Method | mAP | $AP^M$ | $AP^L$ | mAR |
|---|---|---|---|---|
| HRNet Sun et al. (2019) | 74.4 | 70.8 | 81.0 | 79.8 |
| +Pose Only (Ours) | **74.8** | **71.3** | **81.1** | 79.8 |
| +Pose&Visual (offline) | 74.0 | 70.7 | 80.4 | 79.8 |
| +Pose&Visual (online) | 74.3 | 71.0 | 80.7 | 79.8 |

In object detection, there are several methods that could be used in ranking. Particularly AP-Loss and DR-loss Chen et al. (2020); Qian et al. (2020). Particularly, AP-loss is to reformulate a pairwise ranking problem as a listwise ranking problem. Given a set of $N$ items $s_1, s_2, .., s_N$ to be ranked, and defining the pairwise difference as $d_{ij} = -(s_i - s_j)$, AP-loss aims to train the model to learn a probability distribution $\mathbf{P}_i$ over the relative ranking of the i-th item with respect to all others, as shown below,

$$\mathbf{P}_i = [\frac{H(d_{i1})}{1 + \sum_{j \neq i} H(d_{i1})}, ...., \frac{H(d_{iN})}{1 + \sum_{j \neq n} H(d_{iN})}]. \tag{12}$$

$H$ is chosen as a sigmoid function, then the formulation of $\mathbf{P}$ reduces to a simple softmax over $j$. The distribution-based ranking approach works well for the object detection because the number of ranked items is fixed as the number of classes (i.e. a fixed $N$). However, in HPE, the poses are infinite as the coordinates are real-valued. The distribution becomes infinitely dense, making it impractical for neural networks to learn a reasonable pattern. Thus, the performance of AP-loss are naturally worse then our method in terms of HPE.

DR-Loss enforces that every foreground (positive) proposal scores higher than every background (negative) proposal by a fixed margin. Assuming there are $K$ negative samples in $N$ items (commonly $K >> N - K$) and assigning $N$ scores $s_1, s_2, ..., s_N$ for them, DR-Loss sets the least margin $\gamma$ and requires $s_p - s_n > \gamma$. The neural networks are naturally optimized by the hinge term,

$$l(s_p, s_n, \gamma) = [s_p - s_n + \gamma]_+, \tag{13}$$

where [] indicates a ReLU function. This also works when the $N$ is finite. For HPE, requiring a least margin drives $s$ to extrems (0 or 1), collapsing the useful range $(1 - \gamma)$. Consequently, DR-Loss performs poorly in this setting.

## F  DO WE NEED EXPLICIT VISUAL FEATURES?

In this section, we verify our argument that once the original score, computed from the model's heatmaps, is provided as input, the contribution of visual features is already distilled into this score, so re-feeding raw visual features is redundant for post-hoc re-ranking. Standard baselines generate heatmaps from visual evidence and set pose confidence as the mean of per-keypoint peak responses; we therefore refer to the original score as the visual score to emphasize this provenance.

Figure 5 shows two cases with high localization quality (OKS = 0.95) that nevertheless receive depressed visual scores. Occluded keypoints (e.g., the upper subject's left foot; red box) yield diffuse

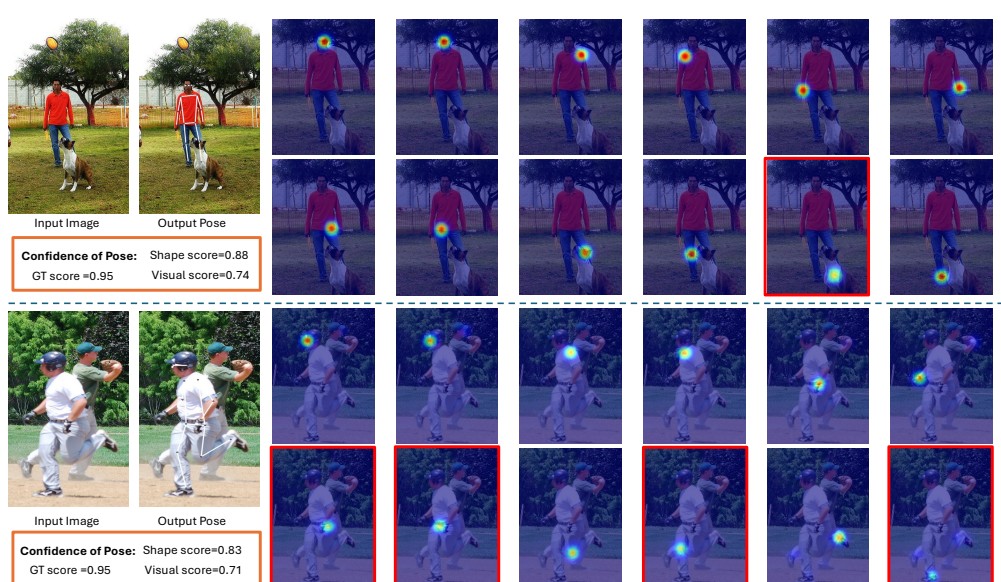

Figure 5: Two representative examples of how confidence scores change with visual evidence. The orange panel reports the ground-truth (GT) confidence, our shape score, and the visual score, computed as the mean of the per-keypoint peak responses of the heatmaps on the right (Notice that visual score also serves as the module input). The red panel highlights keypoints where heatmaps are unreliable due to occlusion or overlap, demonstrating the shape score aligns better with GT than the visual score.

heatmaps with lower peaks. For the visible keypoints, the appearances (e.g., the lower subject's footwear) is also possible to reduce peak responses due to data imbalance (e.g., this particular shoe is not shown up in the training set). Appearance cues are essential for forming the visual score, but injecting them a second time at tends to re-introduce the same biases where our process should correct them.

Quantitatively, on HRNet (Table 9), adding explicit visual features to our re-scoring module produces only a marginal change (e.g., $74.4 \rightarrow 74.0$), consistent with the hypothesis that the visual score already summarizes the relevant appearance information. The small drop likely reflects instances where emphasizing raw features again misleads the model toward the misleading cues illustrated in Figure 5.

In a summary, explicit visual features substantially increase complexity with little gain. HRNet's feature map is roughly 480, 64, 48, orders of magnitude larger than the pose coordinates (17, 2) and a scalar visual score, inflating parameters and memory. Given that the visual score already captures appearance evidence, while additional features add compute and can harm robustness in occluded/rare cases, we omit raw visual features from our re-scoring design.

## G  PROBABILITY FORECAST: GAUSSIAN OR LAPLACIAN?

In the main paper, we instantiate the rank loss under Laplacian assumptions. A natural question is why we do not adopt a Gaussian model instead. This section clarifies the rationale behind our choice.

Our initial motivation for the Laplacian assumptions is the work on probability forecasting, where the objective is to design a proper scoring rule that maps each probabilistic prediction to a value in [0,1]. Consider weather forecasting: a model might report a 70% chance of rain and a 30% chance of sun, and the forecaster needs a principled score to judge how well that distribution matches the actual outcome (observation). Our pose estimation setting is analogous—we also assign a confidence score to each prediction. So, it is natural to borrow insights from some proper scoring rules.

In probability forecast, the key is to design a proper scoring rule. Assume a forecaster make a prediction of distribution $D$ for an event $q$ sampled from a true distribution $Q$. A score rule $S(D, q)$ assigns a numerical reward to that prediction, and the expected value is $S(D, Q) = \mathbb{E}_Q[S(D, q)]$. Ideally, we encourage the forecaster to report the truth, (i.e. we expect the the predicted distribution $D$ matches the true distribution $Q$). Therefore, a strictly proper (negatively-oriented) scoring rule must satisfy,

$$S(Q, Q) \leq S(D, Q),$$

with equality if and only if $D = Q$.

Although many strictly proper rules exist, several rely on evaluating full probability densities, which is often impractical in real-world applications. By contrast, the Continuous Ranked Probability Score (CRPS) is expressed in terms of Cumulative Distribution Functions (CDFs), allowing it to be computed without probability densities and thus making it practical for many real-world applications. The CRPS is

$$\text{CRPS}(F, q) = \int_{-\infty}^{\infty} (F(x) - \mathcal{I}\{x \geq q\})^2 dx,$$

where $\mathcal{I}$ is the indicator function and $F$ is the CDF of the predicted distribution. Intuitively, CRPS measures the squared gap between the predicted CDF and the empirical CDF defined by the single observed outcome $q$ (i.e. $\mathcal{I}\{y \geq q\}$). Fortunately, Baringhaus et al. Baringhaus & Franz (2004) and Rizzo et al. Székely & Rizzo (2005) derived the following closed-form,

$$\text{CRPS}(F, q) = \mathbb{E}_F[|X - q|] - \frac{1}{2}\mathbb{E}_F[|X - X'|],$$

where $X$ and $X'$ are i.i.d. random variables with the CDF $F$.

In practice, we learn the predictive distribution that minimizes CRPS. We parameterize the CDF as $F_\theta$ and use the CRPS directly as a loss function:

$$\text{CRPS}(\theta) = \mathbb{E}[|X_\theta - q|] - \frac{1}{2}\mathbb{E}[|X_\theta - X_\theta'|].$$

When apply to our problem setting, the CRPS matches $s' = \mu = f_\theta$ with $s^*$, obtaining

$$\text{CRPS}(\theta) = \mathbb{E}_i[|f_\theta^i - s_i^*|] - \frac{1}{2}\mathbb{E}_{ij}[|\delta_\theta^{ij}|].$$

Compare with our rank loss with a proper choice of the hyperparameters,

$$l(\theta) = \mathbb{E}_i[|s_i^* - f_\theta^i|] + \frac{1}{2}\mathbb{E}_{ij}[g(\delta_\theta^{ij})],$$

where,

$$g(\delta_\theta^{ij}) = \begin{cases} g_\prec = (|\delta_\theta^{ij}| - \frac{1}{2})^2 & \text{if } \prec \\ g_{\nprec} = |\delta_\theta^{ij}| & \text{if } \nprec \end{cases} \tag{14}$$

one immediately observes that the first term is identical. In fact, if we assume all ranking predictions are correct, the second term $g(\delta_\theta^{ij}) = g_\prec$. Then, the rank loss can be rearranged as

$$l(\theta) \propto \mathbb{E}_i(|s_i^* - f_\theta^i|) - \frac{1}{2}\mathbb{E}_{ij}[|\delta_\theta^{ij}|] + \frac{1}{2}E[(\delta_\theta^{ij})^2].$$

The only extra term in our rank loss is the regularizer $E_{ij}[(\delta_\theta^{ij})^2]$. This penalty prevents the margin $m_{ij} = |\delta_\theta^{ij}|$ from becoming excessively large, preserving score range for other poses.

Hence our Laplacian-form rank loss can be viewed as a rank-aware CRPS: if the pairwise ordering is not correct, it secures. If correct, it acts as a regularized CRPS that aligns the predicted confidence distribution $s'$ with the ground truth $s^*$.

This analysis highlights a key advantage of the Laplacian loss: its clear, CRPS-based interpretation makes the training objective easy to explain. By contrast, the Gaussian variant offers no comparable link and is therefore less interpretable. We present the Gaussian-form rank loss as follow,

$$l_{Guass}(\theta) = \frac{2}{\sigma^2}\mathbb{E}_i[(s^* - f_\theta^i)^2] + \frac{1}{6(\sigma')^2}\mathbb{E}_{ij}[g'(\delta^{ij\theta})],$$

Table 10: Ablation study on different loss form on COCO val set. mAP is reported. Best is **bolded**.

| Method | Ori. | CRPS | Gaussian | Ours (Laplacian) |
|---|---|---|---|---|
| HRNet Sun et al. (2019) | 74.4 | 69.5 | 74.6 | **74.7** |
| ViTPose Xu et al. (2022) | 75.8 | 69.8 | 75.9 | **76.0** |
| GroupPose Liu et al. (2023b) | 72.1 | 62.5 | 72.3 | **72.4** |
| EDPose Yang et al. (2023) | 71.6 | 61.9 | 71.9 | **72.0** |
| DEKR Geng et al. (2021) | 67.1 | 67.9 | 68.6 | **69.0** |
| PINet Wang et al. (2021) | 67.4 | 68.3 | 68.9 | **69.2** |

where,

$$g(\delta_\theta^{ij}) = \begin{cases} g_> = (\delta_\theta^{ij} - c)^3 & \text{if } s_i^* \geq s_j^* \\ g_< = (\delta_\theta^{ij} + c)^3 & \text{if } s_i^* < s_j^* \end{cases} \qquad (15)$$

The Gaussian-form rank loss is brutally straightforward: whenever a pair should satisfy $s_i' > s_j'$, it just maximizes the margin $m_{ij}$ (and vice-versa). Because it lacks the regularization term present in the Laplacian-form loss, it keeps pushing the confidence to a extreme value, hurting the overall performance.

Experimentally, we show the results of alternative loss forms on Tab. 10. The Laplacian form(ours) yields the highest mAP on every baseline, beating the Gaussian loss by 0.1–1.9 points and surpassing CRPS by an even larger margin. CRPS degrades performance on the four top-down models and yields only minor gains on the two bottom-up ones because its second term merely pushes the scores apart without enforcing any ranking. The Gaussian loss offers modest improvements over the baselines but still trails our Laplacian variant, which benefits from the additional regularization term. These results confirm that the Laplacian loss delivers the most consistent and reliable boost in pose-estimation accuracy.

## H ADDITIONAL ABLATION STUDIES ON ARCHITECTURES AND CALIBRATION METRICS.

We examine our method in additional two dimensions: architectural design (Tab. 11) and calibration metrics (Tab. 12). In Tab. 11 we replace our self-attention encoder with an MLP-based architecture identical to DEKR's ScoreNet Geng et al. (2021) and evaluate three representative HPE baselines. Across all baselines, the attention-based module outperforms the MLP variant, which confirms that explicitly modeling pairwise interactions via attention is more effective than independent per-pose rescoring. Tab. 12 reports results on two standard calibration measures, Pearson correlation and AUSE-OKS. Even though our loss is not designed to optimize calibration directly, RCNet consistently improves both metrics. This suggests that sharpening the relative ordering of confidence scores also helps align them more closely with their true reliability.

Table 11: Ablation study on different architecture. MLP indicates using MLP-based scorenet in Geng et al. (2021). Attn. indicates use our attention-based RCNet.

| Method | mAP | $AP^M$ | $AP^L$ | mAR |
|---|---|---|---|---|
| HRNet Sun et al. (2019) | 74.4 | 70.8 | 81.0 | 79.8 |
| +MLP | 74.5 | 71.0 | 81.0 | 79.8 |
| +Attn. | 74.8 | 71.1 | 81.1 | 79.8 |
| EDPose Yang et al. (2023) | 71.6 | 66.2 | 79.7 | 79.3 |
| +MLP | 71.4 | 66.4 | 79.2 | 79.3 |
| +Attn. | 72.0 | 66.7 | 80.0 | 79.3 |
| DEKR Geng et al. (2021) | 67.1 | 61.5 | 76.1 | 72.9 |
| +MLP. | 68.3 | 62.4 | 77.6 | 72.9 |
| +Attn. | 68.9 | 62.6 | 78.6 | 72.9 |

Table 12: Ablation study on different metric of calibration. Pearson Corr. is the Pearson correlation between predicted confidence and empirical accuracy, indicating linear alignment. AUSE-OKS is the area under the sparsification-error curve; lower values mean the uncertainty scores more effectively rank poorly estimated poses.

| Method | Pearson Corr. ↑ | AUSE-OKS (%)↓ |
|---|---|---|
| HRNet Sun et al. (2019) | 0.414 | 1.42 |
| +Ours | 0.547 | 1.31 |
| EDPose Yang et al. (2023) | 0.430 | 1.98 |
| +Ours | 0.446 | 1.87 |
| DEKR Geng et al. (2021) | 0.241 | 2.61 |
| +Ours | 0.326 | 1.03 |

## I  ADDITIONAL QUALITATIVE SAMPLES

Below, we provide more qualitative samples.

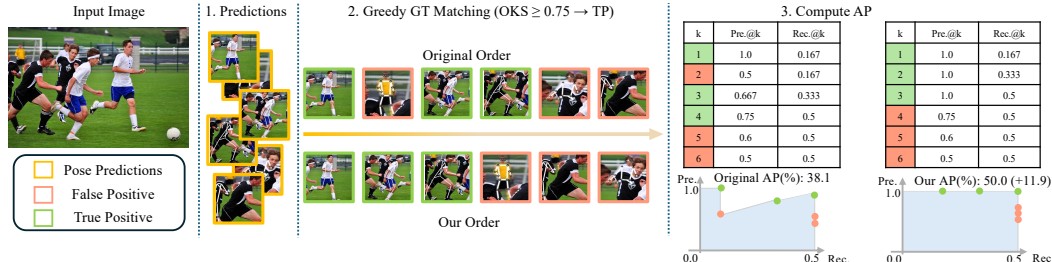

Figure 6: A representative qualitative example of ranking correction on COCO val. From left to right: input image; predictions from the HPE model; COCO's greedy matching after sorted by confidence (with false positives in pink and true positives in green); and the revised ordering showing improved AP for this image.

## J  PROOF

### J.1  PROOF OF THM.1

We aim to learn the model parameters $\theta$ by maximizing the expected log probability of the predicted scores' ordering being consistent with the ground truth ordering (equivalently, minimizing $L(\theta)$):

$$
\begin{aligned}
\theta^* &= \arg\min_{\theta} L(\theta) \\
&= \arg\min_{\theta} -\mathbb{E}_{ij}[\log \mathbf{P}_\theta(s_i' \prec s_j', s_i^* \prec s_j^*)] \\
&= \arg\min_{\theta} -\mathbb{E}_{ij}[\log \mathbf{P}_\theta(s_i' \prec s_j'|s_i^*, s_j^*)p(s_i^*)p(s_j^*)] \\
&= \arg\min_{\theta} -2\mathbb{E}_i[\log p(s_i^*)] - \mathbb{E}_{ij}[\log \mathbf{P}_\theta(s_i' \prec s_j'|s_i^*, s_j^*)] \\
&= \arg\min_{\theta} -\mathbb{E}_{ij}[\log \mathbf{P}_\theta(s_i' \prec s_j'|s_i^*, s_j^*)] \\
&= \arg\min_{\theta} -\mathbb{E}_{ij}[\log \mathbf{P}_\theta(s_i' \prec s_j'|s_i^*, s_j^*, \mu_i, \mu_j)p_\theta(\mu_i|s_i^*)p_\theta(\mu_j|s_j^*)] \\
&= \arg\min_{\theta} -\mathbb{E}_{ij}[\log \mathbf{P}_\theta(s_i' \prec s_j'|s_i^*, s_j^*, \mu_i, \mu_j)] - \mathbb{E}_{ij}[\log p_\theta(\mu_i|s_i^*)] - \mathbb{E}_{ij}[\log p_\theta(\mu_j|s_j^*)] \\
&= \arg\min_{\theta} -\mathbb{E}_{ij}[\log \mathbf{P}_\theta(s_i' \prec s_j'|s_i^*, s_j^*, \mu_i, \mu_j)] - \mathbb{E}_i[\log p_\theta(\mu_i|s_i^*)] - \mathbb{E}_j[\log p_\theta(\mu_j|s_j^*)] \\
&= \arg\min_{\theta} -2\mathbb{E}_i[\log p_\theta(\mu_i|s_i^*)] - \mathbb{E}_{ij}[\log \mathbf{P}_\theta(s_i' \prec s_j'|s_i^*, s_j^*, \mu_i, \mu_j)]
\end{aligned}
\tag{16}
$$

In the main paper, $\mu$ is parameterized as a neural network $\mu_\theta$. Additionally, we introduce two assumptions: (1) the predicted score $s_i'$ is modeled as a Laplacian variable centered at $\mu_\theta(s_i, \boldsymbol{p}_i)$, denoted as $\mu_\theta^i$ for simplicity, and (2) the $\mu_\theta^i$ is Laplace-distributed around $s^*$. Under these assumptions, we obtain the following equation:

$$
\begin{aligned}
\theta^* &\propto \arg\min_{\theta} 2\mathbb{E}_i\left[\frac{|s_i^* - \mu_\theta^i|}{b'}\right] - \mathbb{E}_{ij}\left[\log \mathbf{P}_\theta(s_i' \prec s_j' \mid s_i^*, s_j^*, \mu_\theta^i, \mu_\theta^j)\right] \\
&= \arg\min_{\theta} \frac{2}{b'}\mathbb{E}_i\left[|s_i^* - \mu_\theta^i|\right] - \mathbb{E}_{ij}\left[\log \mathbf{P}_\theta(s_i' \prec s_j' \mid s_i^*, s_j^*, \mu_\theta^i, \mu_\theta^j)\right].
\end{aligned}
\tag{17}
$$

We define $\overline{g}(\delta_\theta^{ij}; s_i^*, s_j^*, \mu_\theta^i, \mu_\theta^j) = -\log \mathbf{P}_\theta(s_i' \prec s_j' \mid s_i^*, s_j^*, \mu_\theta^i, \mu_\theta^j)$, where $\delta_\theta^{ij} = \mu_\theta^i - \mu_\theta^j$, so that,

$$
\theta^* \propto \arg\min_{\theta} \frac{2}{b'}\mathbb{E}_i\left[|s_i^* - \mu_\theta^i|\right] + \mathbb{E}_{ij}\left[\overline{g}(\delta_\theta^{ij}; s_i^*, s_j^*, \mu_\theta^i, \mu_\theta^j)\right].
\tag{18}
$$

We denote the difference between two refined scores $s_i', s_j'$ as $z_{ij}$, where $z_{ij} \in [-c, c]$. Since $s_i' \prec \text{Laplace}(\mu_\theta^i, b)$ and $s_j' \prec \text{Laplace}(\mu_\theta^j, b)$ are independent, $z_{ij} = s_i' - s_j'$ is given by an subtraction of two Laplacian distributions:

$$p_\theta(z_{ij}) = \frac{1}{4b^2} \left( b + \left| z_{ij} - \delta_\theta^{ij} \right| \right) \exp\left( -\frac{\left| z_{ij} - \delta_\theta^{ij} \right|}{b} \right). \tag{19}$$

The $\bar{g}$ in a form of $z_{ij}$ is shown below,

$$\begin{aligned}
\bar{g}(\delta_\theta^{ij}; s_i^*, s_j^*, \mu_i, \mu_j) &= -\log \mathbf{P}_\theta(s_i' \prec s_j' \mid s_i^*, s_j^*, \mu_i, \mu_j)) \\
&= -\log \mathbf{P}_\theta(s_i' - s_j' \prec 0 \mid s_i^*, s_j^*, \mu_i, \mu_j)) \\
&= -\log \mathbf{P}_\theta(z_{ij} \prec 0 \mid s_i^*, s_j^*, \mu_i, \mu_j)) \\
&= -\log \int_{-c}^{c} p_\theta(z_{ij} \prec 0 \mid s_i^*, s_j^*, \mu_i, \mu_j) dz_{ij}
\end{aligned} \tag{20}$$

To apply Jensen's inequality, we introduce a uniformly-distributed random variable $u \sim U(-c, c)$. Recall that the expectation of a function under a uniform distribution is defined as:

$$\mathbb{E}_{u \sim U(-c,c)}[f(u)] = \frac{1}{2c} \int_{-c}^{c} f(u) \, du. \tag{21}$$

Therefore, the integral is rewritten as:

$$\int_{-c}^{c} p_\theta(z_{ij} \prec 0 \mid s_i^*, s_j^*, \mu_i, \mu_j) \, dz_{ij} = 2c \cdot \mathbb{E}_{u \sim U(-c,c)} \left[ p_\theta(z_{ij} = u \prec 0 \mid s_i^*, s_j^*, \mu_\theta^i, \mu_\theta^j) \right]. \tag{22}$$

Substituting this into the original expression gives:

$$\begin{aligned}
\bar{g}(\delta_\theta^{ij}; s_i^*, s_j^*, \mu_i, \mu_j) &= -\log \left( 2c \cdot \mathbb{E}_{u \sim U(-c,c)} \left[ p_\theta(z_{ij} = b \prec 0 \mid s_i^*, s_j^*, \mu_\theta^i, \mu_\theta^j) \right] \right) \\
&= -\log 2c - \log \mathbb{E}_{u \sim U(-c,c)} \left[ p_\theta(z_{ij} = b \prec 0 \mid s_i^*, s_j^*, \mu_\theta^i, \mu_\theta^j) \right] \\
&\leq -\log 2c - \mathbb{E}_{u \sim U(-c,c)} \left[ \log p_\theta(z_{ij} = b \prec 0 \mid s_i^*, s_j^*, \mu_\theta^i, \mu_\theta^j) \right]
\end{aligned} \tag{23}$$

We can further simplify the equation as follows by removing variables irrelevant to $\theta$,

$$\begin{aligned}
\bar{g}(\delta_\theta^{ij}; s_i^*, s_j^*, \mu_\theta^i, \mu_\theta^j) &\leq -\log 2c - \mathbb{E}_{u \sim U(-c,c)} \left[ \log p_\theta(z_{ij} = u \prec 0 \mid s_i^*, s_j^*, \mu_\theta^i, \mu_\theta^j) \right] \\
&\propto -\mathbb{E}_{u \sim U(-c,c)} \left[ \log p_\theta(z_{ij} = u \prec 0 \mid s_i^*, s_j^*, \mu_\theta^i, \mu_\theta^j) \right] \\
&\propto -\int_{-c}^{c} \log p_\theta(z_{ij} \prec 0 \mid s_i^*, s_j^*, \mu_\theta^i, \mu_\theta^j) \, dz_{ij}.
\end{aligned} \tag{24}$$

Eq. 24 is further expanded as follows.

$$- \int_{-c}^{c} \log p_\theta(z_{ij} \prec 0 \mid s_i^*, s_j^*, \mu_i, \mu_j)) \, dz_{ij} \tag{25}$$

$$= \begin{cases} \int_0^c - \log \left( \frac{1}{4b^2} (b + |z_{ij} - \delta_\theta^{ij}|) \exp\left( -\frac{|z_{ij} - \delta_\theta^{ij}|}{b} \right) \right) dz_{ij}, & \text{if } s_i^* \geq s_j^* \\ \int_{-c}^0 - \log \left( \frac{1}{4b^2} (b + |z_{ij} - \delta_\theta^{ij}|) \exp\left( -\frac{|z_{ij} - \delta_\theta^{ij}|}{b} \right) \right) dz_{ij}, & \text{if } s_i^* < s_j^* \end{cases} \tag{26}$$

$$= \begin{cases} \int_0^c \left( \frac{1}{b} |z_{ij} - \delta_\theta^{ij}| - \log(b + |z_{ij} - \delta_\theta^{ij}|) + \log(4b^2) \right) dz_{ij}, & \text{if } s_i^* \geq s_j^* \\ \int_{-c}^0 \left( \frac{1}{b} |z_{ij} - \delta_\theta^{ij}| - \log(b + |z_{ij} - \delta_\theta^{ij}|) + \log(4b^2) \right) dz_{ij}, & \text{if } s_i^* < s_j^* \end{cases} \tag{27}$$

$$\leq \begin{cases} \int_0^c \left( \frac{1}{b} |z_{ij} - \delta_\theta^{ij}| + \log(b) \right) dz_{ij}, & \text{if } s_i^* \geq s_j^* \\ \int_{-c}^0 \left( \frac{1}{b} |z_{ij} - \delta_\theta^{ij}| + \log(b) \right) dz_{ij}, & \text{if } s_i^* < s_j^* \end{cases} \tag{28}$$

$$\propto \begin{cases} \int_0^c \frac{1}{b} |z_{ij} - \delta_\theta^{ij}| \, dz_{ij}, & \text{if } s_i^* \geq s_j^* \\ \int_{-c}^0 \frac{1}{b} |z_{ij} - \delta_\theta^{ij}| \, dz_{ij}, & \text{if } s_i^* < s_j^* \end{cases} \tag{29}$$

We further discuss the relation between $z_{ij}$ and $\delta_\theta^{ij}$ and distinguish the following four cases:

1. $s_i^* > s_j^*, \delta_\theta^{ij} \in [0, c]$:
$$g_1(\delta_\theta^{ij}) = \frac{1}{2}(\delta_\theta^{ij} - c)^2 + \frac{1}{2}(\delta_\theta^{ij})^2$$

2. $s_i^* < s_j^*, \delta_\theta^{ij} \in [0, c]$:
$$g_2(\delta_\theta^{ij}) = \frac{1}{2}(\delta_\theta^{ij} + c)^2 - \frac{1}{2}(\delta_\theta^{ij})^2$$

3. $s_i^* > s_j^*, \delta_\theta^{ij} \in [-c, 0)$:
$$g_3(\delta_\theta^{ij}) = \frac{1}{2}(\delta_\theta^{ij} - c)^2 - \frac{1}{2}(\delta_\theta^{ij})^2$$

4. $s_i^* < s_j^*, \delta_\theta^{ij} \in [-c, 0)$:
$$g_4(\delta_\theta^{ij}) = \frac{1}{2}(\delta_\theta^{ij} + c)^2 + \frac{1}{2}(\delta_\theta^{ij})^2$$

This is exactly the definition of $g(\delta_\theta^{ij})$ and thus $\bar{g} < g$.

Consequently, we prove the Thm.1 as

$$\theta^* = \arg\min_\theta L(\theta)$$

$$= \arg\min_\theta -2\mathbb{E}_i[\log p(\mu_\theta^i | s_i^*)] - \mathbb{E}_{ij}[\log \mathbf{P}_\theta(s_i' \prec s_j' | s_i^*, s_j^*, \mu_\theta^i, \mu_\theta^j)]$$

$$\propto \arg\min_\theta \frac{2}{b'} \mathbb{E}_i \left[ |s_i^* - \mu_\theta^i| \right] + \mathbb{E}_{ij} \left[ \bar{g}(\delta_\theta^{ij}; s_i^*, s_j^*, \mu_\theta^i, \mu_\theta^j) \right] \tag{30}$$

$$\leq \arg\min_\theta \frac{2}{b'} \mathbb{E}_i[|s_i^* - \mu_\theta^i|] + \mathbb{E}_{ij}[g(\delta_\theta^{ij}; s_i^*, s_j^*, \mu_\theta^i, \mu_\theta^j)]$$

$$= \arg\min_\theta l(\delta_\theta^{ij}; s_i^*, s_j^*, \mu_\theta^i, \mu_\theta^j)$$

## J.2 Proof of Prop.2

Considering a pairwise ranking of two poses $p_1$ and $p_2$ by assigning them scores $s_1 \in \mathbb{R}[0, c]$ and $s_2 \in \mathbb{R}[0, c]$. Without the loss of the generosity, assume $0 < s_1 < s_2 < c$. Once $s_1$ is chosen, the optimal margin $m^*$ that maximizes the difference between $s_1$ and $s_2$ is,

$$m^* = c - s_1.$$

With no prior knowledge of $s_1$, it is reasonable to assume $s_1$ is uniformly distributed on $\mathbb{R}[0, c]$. Under this assumption, the expectation of the optimal margin $m^*$ is $\frac{c}{2}$, shown in the following equations.

$$\mathbb{E}[m^*] = \int_0^c m^* p(s_1) ds_1 \tag{31}$$

$$= \int_0^c (c - x) \frac{1}{c} dx \tag{32}$$

$$= \frac{1}{c} \int_0^c (c - x) dx \tag{33}$$

$$= \frac{1}{c}[(cx - \frac{1}{2}x^2)|_{x=c} - (cx - \frac{1}{2}x^2)|_{x=0}] \tag{34}$$

$$= \frac{1}{c}[c^2 - \frac{1}{2}c^2 - 0] \tag{35}$$

$$= \frac{c}{2} \tag{36}$$

## J.3 Proof of Thm.3 and Cor.4

There are a few preparations before the proof. First, we provide the following assumption. This assumption is standard for any well-designed network in regression tasks optimized with $L_1$ or $L_2$-style loss, and is both empirically and theoretically supported by He et. al. He et al. (2015) and Sexe et al, Saxe et al. (2013).

**Assumption 4.** *For a scalar-valued regression network $f_\theta$ initialized with a sufficiently small $\mathrm{Var}[\delta_{\theta_0}^{ij}]$ and proper learning rate $r$, the variance of output dose not decrease and converges to ground truth variance $\mathrm{Var}[\delta^*]$, i.e. $\mathrm{Var}[\delta_{\theta_t}^{ij}] \leq \mathrm{Var}[\delta_{\theta_{t+1}}^{ij}] \leq \mathrm{Var}[\delta^*]$, where $\mathrm{Var}[\delta^*]$ is the variance of the pairwise difference of the ground truth. Further, assuming $\mathbb{E}[(\delta_{\theta_t}^{ij})] = \mathbb{E}[(\delta_{\theta_{t+1}}^{ij})]$, it follows that $\mathbb{E}[(\delta_{\theta_t}^{ij})^2] \leq \mathbb{E}[(\delta_{\theta_{t+1}}^{ij})^2]$, if the inputs $x_i$ and $x_j$ are i.i.d. sampled.*

Second, we index a pair (ij) as $k$ for simplicity. So, $\delta_\theta^k = \delta_\theta^{ij}$ and we use $\prec$ when the relation of $s'$ matches the relation of $s^*$, i.e. if $s_i' > s_j'$ when $s_i^* > s_j^*$ (vice versa), and $\not\prec$ for mismatching (misordered). Additionally, $N$ is used to denote the total number of pairs with $N_\prec$ and $N_{\not\prec}$ for the number of ordered and misordered pairs respectively. The last preparation is to simplify the $g$. Note the $g$ could be re-written in the following form with simple algebraic operations.

$$g(\delta_\theta^k; s_i^*, s_j^*) = \begin{cases} g_1(\delta_\theta^k) = (\delta_\theta^k - \frac{c}{2})^2 + \frac{c^2}{4} & \text{if } \prec \text{ and } \delta_\theta^k \in [0, c] \\ g_2(\delta_\theta^k) = c\delta_\theta^k + \frac{c^2}{2} & \text{if } \not\prec \text{ and } \delta_\theta^k \in [0, c] \\ g_3(\delta_\theta^k) = -c\delta_\theta^k + \frac{c^2}{2} & \text{if } \prec, \text{ and } \delta_\theta^k \in [-c, 0) \\ g_4(\delta_\theta^k) = (\delta_\theta^k + \frac{c}{2})^2 + \frac{c^2}{4} & \text{if } \not\prec, \text{ and } \delta_\theta^k \in [-c, 0) \end{cases} \tag{37}$$

Considering the sign of $\delta_\theta^k$, the $g$ could be further integrate into a even simpler format,

$$g(\delta_\theta^k) = \begin{cases} g_\prec(\delta_\theta^k) = (|\delta_\theta^k| - \frac{c}{2})^2 + \frac{c^2}{4} & \text{if } \prec \\ g_{\not\prec}(\delta_\theta^k) = c|\delta_\theta^k| + \frac{c^2}{2} & \text{if } \not\prec \end{cases} \tag{38}$$

We now start the proof. Notice that our rank loss contains two items. The first item is a L1 loss. The second item is a function w.r.t. the pairwise difference of the output. As we assume the total loss

decease, either one of the items or both decrease. We now show that no matter what item dominate, the ranking accuracy will increase in a certain rate.

For the first term, we introduce the following lemma. The lemma show that if the L1 term decreases, the ranking accuracy $\mathbf{P}$ in a certain iteration $t$ must increase in a rate of $o(\frac{2}{\min_k |\delta_k^*|} \mathbb{E}_i[|\mu_t^i - s_i^*|])$, where $\mu_t^i$ is the $\mu_\theta^i$ at iteration $t$.

**Lemma 5.** $\mathbf{P}_{correct}^t \geq 1 - \frac{2}{\min_k |\delta_k^*|} \mathbb{E}_i[|\mu_t^i - s_i^*|]$.

The proof the Lemma. 5 start with the investigation of a misordered pair $(i, j)$. With out the loss of generality, assumes $s_i^* > s_j^*$ yet $\mu_\theta^i < \mu_\theta^j$ in the iteration $t$. Then

$$s_i^* - s_j^* = (\mu_t^i - \mu_t^j) + [(s_i^* - \mu_t^i) - (s_j^* - \mu_t^j)]. \tag{39}$$

Since $\mu_t^i < \mu_t^j$,

$$s_i^* - s_j^* < 0 + (s_i^* - \mu_t^i) - (s_j^* - \mu_t^j) < |(s_i^* - \mu_t^i) - (s_j^* - \mu_t^j)| < |s_i^* - \mu_t^i| + |s_j^* - \mu_t^j|. \tag{40}$$

Thus, a misordered pair satisfies that,

$$|s_i^* - \mu_t^i| + |s_j^* - \mu_t^j| > s_i^* - s_j^* > |s_i^* - s_j^*| > \min_k |\delta_k^*|, \tag{41}$$

where $\delta_k^* = s_i^* - s_j^*$. Suppose $N_{\not\prec}$ is the number of misordered pairs, summing inequality gives,

$$N_{\not\prec} \min_k |\delta_k^*| \leq \sum_{\text{misordered}} (|s_i^* - \mu_t^i| + |s_j^* - \mu_t^j|) \leq \sum_{\text{all pairs}} (|s_i^* - \mu_t^i| + |s_j^* - \mu_t^j|) = 2n \sum |s_i^* - \mu_t^i|. \tag{42}$$

Therefore,

$$N_{\not\prec} \leq \frac{2n}{\min_k |\delta_k^*|} \sum |s_i^* - \mu_t^i|. \tag{43}$$

Finally, divided by the total number of pairs, we obtain

$$1 - \mathbf{P}_{correct}^t \leq \frac{2}{\min_k |\delta_k^*|} \frac{1}{n} \sum |s_i^* - \mu_t^i| = \frac{2}{\min_k |\delta_k^*|} \mathbb{E}_i[|s_i^* - \mu_t^i|]. \tag{44}$$

Thus, we complete the proof of Lemma. 5

$$\mathbf{P}_{correct}^t \geq 1 - \frac{2}{\min_k |\delta_k^*|} \mathbb{E}_i[|s_i^* - \mu_t^i|] \tag{45}$$

We then show behavior of the second term in rank loss $g(\delta_\theta^{ij})$ also directly connects to the rank accuracy $\mathbf{P}_{correct}$. The target is to show the decrease of $g$ leads to the increase of ranking accuracy only. We firstly define following sets.

**Definition 2.** $A = \{(ij) \mid (ij) \in ([0, N] \times [0, N])\}$ *is the index set for all the pairs in the dataset. Let* $N_A = |A|$ *is the total number of elements in A. There are four subsets of A.* $A_{\prec \to \not\prec}$ *is the index set of pairs that contains correct relation in t-th iteration, yet changed to wrong relation after one step optimization (similarly for* $A_{\not\prec \to \prec}$*).* $A_{\prec}$ *is the index set of pairs that is still correct in the* $(t+1)$*-th iteration (*$A_{\not\prec}$ *vice versa).*

For simplicity, we use $A_1$, $A_2$, $A_3$ and $A_4$ to denote $A_{\prec \to \not\prec}$, $A_{\not\prec \to \prec}$, $A_{\prec}$ and $A_{\not\prec}$ respectively in the following proof. Additionally, we define $\bar{\Delta}_k = \delta_{t+1}^k - \delta_t^k$ as the deviation of $\delta$ of the $k$-th pair. For each subset, we analyze them item by item.

For $D_{A_1}$,

$$D_{A_1} = \frac{1}{N_A} \sum_{A_1} [g_{\nprec} - g_{\prec}] \tag{46}$$

$$= \mathbb{E}[c|\delta_{t+1}^k| + \frac{c^2}{2} - (|\delta_t^k| - \frac{c}{2})^2 - \frac{c^2}{4}] \tag{47}$$

$$= \frac{1}{N_A} \sum_{A_1} [c|\delta_{t+1}^k| - (\delta_t^k)^2 + c|\delta_t^k|] \tag{48}$$

$$= \frac{1}{N_A} \sum_{A_1} [c(|\delta_{t+1}^k| + |\delta_t^k|) - (\delta_t^k)^2] \tag{49}$$

$$= \frac{1}{N_A} \sum_{A_1} [c(|\delta_t^k + \bar{\Delta}_k| + |\delta_t^k|) - (\delta_t^k)^2] \tag{50}$$

$$= \frac{1}{N_A} \sum_{A_1} [c|\bar{\Delta}_k| - (\delta_t^k)^2] \geq 0. \tag{51}$$

From Equ. 36 to Equ. 37, we exploit the fact that $\delta_t^k$ and $\delta_{t+1}^k$ should have different sign (so is $\bar{\Delta}_k$ and $\delta_t^k$) as $A_1$ indicates the pair moves from the correct relation to wrong relation. **It worth notice that $D_{A_1} \geq 0$ because $|\delta_t^k| < |\bar{\Delta}_k| \leq c$.** Take $N_A \to \infty$,

$$D_{A_1} = \mathbf{P}(A_1)\mathbb{E}[c|\bar{\Delta}| - (\delta_t)^2] > 0. \tag{52}$$

Similarly for $D_{A_2}$,

$$D_{A_2} = \mathbf{P}(A_2)\mathbb{E}[(\delta_{t+1})^2 - c|\bar{\Delta}|] < 0. \tag{53}$$

For $D_{A_3}$,

$$D_{A_3} = \frac{1}{N_A} \sum_{A_3} [(|\delta_{t+1}^k| - \frac{c}{2})^2 - (|\delta_t^k| - \frac{c}{2})^2] \tag{54}$$

$$= \frac{1}{N_A} \sum_{A_3} [(\delta_{t+1}^k)^2 - c|\delta_{t+1}^k| - (\delta_t^k)^2 + c|\delta_t^k|] \tag{55}$$

$$= \frac{1}{N_A} \sum_{A_3} [(\delta_{t+1}^k)^2 - (\delta_t^k)^2 + c(|\delta_t^k| - |\delta_{t+1}^k|)] \tag{56}$$

$$= \frac{1}{N_A} \sum_{A_3} [(\delta_{t+1}^k)^2 - (\delta_t^k)^2] + \frac{1}{N_A} \sum_k [c(|\delta_t^k| - |\delta_{t+1}^k|)] \tag{57}$$

$$= \frac{1}{N_A} \sum_{A_3} [(\delta_{t+1}^k)^2 - (\delta_t^k)^2] - \frac{N_{\delta_t^k > 0}}{N_A} \frac{1}{N_{\delta_t^k > 0}} \sum_{\delta_t^k > 0} [c|\bar{\Delta}_k|] + \frac{N_{\delta_t^k < 0}}{N_A} \frac{1}{N_{\delta_t^k < 0}} \sum_{\delta_t^k < 0} [c|\bar{\Delta}_k|] \tag{58}$$

Similarly, take $N_A \to \infty$,

$$D_{A_3} = \mathbf{P}(A_3)\mathbb{E}[(\delta_{t+1}^k)^2 - (\delta_t^k)^2] - \mathbf{P}(\delta_t^k > 0)\mathbb{E}[c|\bar{\Delta}_k|] + \mathbf{P}(\delta_t^k < 0)\mathbb{E}[c|\bar{\Delta}_k|] \tag{59}$$

$$= \mathbf{P}(A_3)\mathbb{E}[(\delta_{t+1}^k)^2 - (\delta_t^k)^2] + [\mathbf{P}(\delta_t^k < 0) - \mathbf{P}(\delta_t^k > 0)]\mathbb{E}[c|\bar{\Delta}_k|]. \tag{60}$$

As we sample each pair independently, $\mathbf{P}(\delta_t^k < 0) - \mathbf{P}(\delta_t^k > 0) = 0.5 - 0.5 = 0$. Therefore,

$$D_{A_3} = \mathbf{P}(A_3)\mathbb{E}[|(\delta_{t+1}^k)^2 - (\delta_t^k)^2] \geq 0, \tag{61}$$

which is greater than 0 based on Assumption 4.

For $D_{A_4}$,

$$D_{A_4} = \frac{1}{N_A} \sum_{A_4} [|c|\delta_{t+1}^k| - c|\delta_t^k||], \tag{62}$$

$$= \frac{1}{N_A} \sum_{A_4} [c(|\delta_t^k + \bar{\Delta}_k| - |\delta_t^k|)]. \tag{63}$$

Take $N_A \to \infty$,

$$D_{A_4} = [\mathbf{P}(\delta_t^k < 0) - \mathbf{P}(\delta_t^k > 0)]\mathbb{E}[c|\bar{\Delta}_k|] \tag{64}$$

$$= 0. \tag{65}$$

Combing all,

$$D = D_{A_1} + D_{A_2} + D_{A_3} + D_{A_4} \tag{66}$$

$$= \mathbf{P}(A_1)\mathbb{E}[c|\bar{\Delta}| - (\delta_t)^2] + \mathbf{P}(A_2)\mathbb{E}[(\delta_{t+1}^k)^2 - c|\bar{\Delta}|] + \mathbf{P}(A_3)\mathbb{E}[(\delta_{t+1}^k)^2 - (\delta_t^k)^2] + 0 \tag{67}$$

$$= (\mathbf{P}(A_1) - \mathbf{P}(A_2))\mathbb{E}[c|\bar{\Delta}| - (\delta_t^k)^2] + \mathbf{P}(A_3)\mathbb{E}[(\delta_{t+1}^k)^2 - (\delta_t^k)^2]. \tag{68}$$

As the loss decreases, $D \le 0$. Therefore,

$$D = (\mathbf{P}(A_1) - \mathbf{P}(A_2))\mathbb{E}[c|\bar{\Delta}| - (\delta_t^k)^2] + \mathbf{P}(A_3)\mathbb{E}[|(\delta_{t+1}^k)^2 - (\delta_t^k)^2] \le 0 \tag{69}$$

$$\Rightarrow (\mathbf{P}(A_1) - \mathbf{P}(A_2))\mathbb{E}[c|\bar{\Delta}| - (\delta_t^k)^2] \le -\mathbf{P}(A_3)\mathbb{E}[|(\delta_{t+1}^k)^2 - (\delta_t^k)^2] \tag{70}$$

$$\Rightarrow \mathbf{P}(A_1) - \mathbf{P}(A_2) \le -\mathbf{P}(A_3)\frac{\mathbb{E}[(\delta_{t+1}^k)^2 - (\delta_t^k)^2]}{\mathbb{E}[c|\bar{\Delta}| - (\delta_t^k)^2]} \tag{71}$$

$$\Rightarrow \mathbf{P}(A_2) \ge \mathbf{P}(A_1) + \mathbf{P}(A_3)\frac{\mathbb{E}[(\delta_{t+1}^k)^2 - (\delta_t^k)^2]}{\mathbb{E}[c|\bar{\Delta}| - (\delta_t^k)^2]} \tag{72}$$

Given that $\mathbf{P}(A_3) \ge 0$, $\mathbb{E}[(\delta_{t+1}^k)^2 - (\delta_t^k)^2] \ge 0$ (by the Assumption 5) and $\mathbb{E}[c|\bar{\Delta}| - (\delta_t^k)^2] > 0$, then

$$\mathbf{P}(A_2) \ge \mathbf{P}(A_1),$$

indicating there must have more pairs become correctly related after one iteration. Consequently,

$$\mathbf{P}_{\text{correct}}^{t+1} = \mathbf{P}_{\text{correct}}^t + (\mathbf{P}(A_2) - \mathbf{P}(A_1)) \ge \mathbf{P}_{\text{correct}}^t. \tag{73}$$

Additionally, assuming a random assignment at the initialization of the model (i.e. $\mathbf{P}_{A_3}^0 = \frac{1}{2}$), the accuracy increasing at least in a scale of $o(\mathbb{E}[(\delta_{t+1}^k)^2 - (\delta_t^k)^2])$, shown by Equ.74.

In concluded, if loss decreases, one of the items must decrease. This leads the $\mathbf{P}_{\text{correct}}$ either increase with rate $o(\mathbb{E}_i[|s_i^* - \mu_t^i|])$ if the first term dominates or $o(\mathbb{E}[(\delta_t^k)^2 - (\delta_{t-1}^k)^2])$ if the second term dominates. Thus the increasing rate is at least within the level of $o(\min\{\mathbb{E}_i[|s_i^* - \mu_t^i|], \mathbb{E}[(\delta_t^k)^2 - (\delta_{t-1}^k)^2]\})$. So we complete the proof of Thm.3.

A direct corollary brought Thm.3 is shown as follow. Although the above corollary assumes additional smoothness and bounded-gradient conditions to guarantee gradient decedent reliably decreases the loss, these requirements are mild and are generally met by mainstream deep neural networks modules (e.g. attention, ReLU or linear layers).

**Corollary 6** (Convergence of Algorithm). *Any neural networks $\mu_\theta$ trained by $l(\theta)$ that satisfies the following properties is bound to converge.*

- *The network $\mu_\theta$ is Gradient Lipchitz Continuous (GLC-condition).*

- *The network $\mu_\theta$ is properly initialized (such as kaiming uniform initialization He et al. (2015)) with a sufficiently small output variance $\mathrm{Var}[\delta_{\theta_0}]$ at the initial step.*

- *The network $\mu_\theta$ is optimized by Gradient Descent (GD) with a proper small learning rate $r$.*

- *Dataset is sufficient large.*

For Cor.6, the only issue is how to guarantee the loss is decreasing. If the loss is decreasing in every iteration, then the ranking accuracy is increasing and bounded. Therefore, it converges. The below lemma that follow directly from the GLC property validates this. The proof is omitted because it is simply applying property.

**Lemma 7.** *For an GLC-network $f_\theta$ optimized by gradient descent, denote $r$ as the learning rate used from $(t)$-th iteration to $(t + 1)$-th iteration, then $\lim_{r \to 0} D = l_{t+1} - l_t = 0^-$., i.e. $l_{t+1} \le l_t$ with a sufficiently small $r$.*

Therefore, we can concluded the proof for Cor.6.

