# OpenReview forum: "On the Ranking of Human Pose Confidence"
_ICLR.cc/2026/Conference — ICLR 2026 Conference Desk Rejected Submission_

### Official Review · Reviewer_TScq · 2025-10-18

**Soundness:** 3
**Presentation:** 3
**Contribution:** 4
**Rating:** 6
**Confidence:** 4

**Summary:**

This paper addresses the neglected problem of ranking confidence scores in 2D human pose estimation (HPE). They propose a novel approach that explicitly optimizes these rankings without altering the keypoint coordinates themselves. The core of their method formulates confidence ranking as a pairwise ordering problem and introduces a rank loss that theoretically guarantees improved rankings by upper-bounding the negative likelihood of incorrect orderings. To validate this, they develop the Ranked Confidence Net (RCNet), a lightweight, post-hoc module that refines only the confidence scores. Experiments on COCO and CrowdPose demonstrate consistent and sometimes substantial improvements

**Strengths:**

1. This paper propose and address a truely existing problem in human pose estimation.
2. This paper have adequate theoretical analysis to illustrate their problem and method.
3. This paper could achieve consistent performance gains on a wide range baselines.
4. This paper introduce a lightweight plug-and-play RCNet with only 0.07M params to refine the pose confidence ranking.

**Weaknesses:**

- Minor
1. Actually on most baseline, the performance gains seem weak, such as HRNet (+0.4), ViTPose (+0.3), GroupPose (+0.3), EDPose (+0.4) and LogoCap (+0.5), and only in baselines which achieve poor mAP could get obvious improvement, like DEKR (+1.9), PINet (+1.8).
2. Their ranking methods could only improve the accuracy in multi person pose estimation, not for single person.
3. Some citation latex code (\cite and \citep) do not use correctly.
4. In figure 1, the caption order do not match the figure.

**Questions:**

1. Besides Laplacian and Gaussian distribution, may there be other assumptions?

---

> ### Author Response · Authors · 2025-11-21
>
> **[W1] Improvements Seems to be Weak.**
> >**TL;DR** Quantitative comparison shows that our lightweight RCNet achieves far higher accuracy-per-cost efficiency than scaling input size or backbone parameters, while consistently improving a wide range of HPE baselines. This highlights mis-ranking as a key overlooked error source which alone accounts for the remaining 15–18 mAP gap even with perfect localization.
>
>
> We believe that determining whether the improvement is marginal is best guided by quantitative evidence rather than subjective impression. Thus, we quantitatively compare our method with the two standard ways top-down HPE improves accuracy: scaling backbone size or input resolution, as Reviewer PMss suggested. These methods increase mAP only at the cost of large extra parameters or GFLOPs, whereas RCNet achieves +0.3–0.4 mAP with negligible computational overhead, achieving far higher mAP improvement per parameter and per FLOP than backbone or input scaling.
>
> **Backbone scaling (same input size)**
>
> | Method | Input  | Backbone Pair     | ΔParams (M) | ΔmAP | mAP Gain / M Params |
> |--------|--------|-------------------|-------------|------|----------------------|
> | Simple Baseline[1] | 256×192   | ResNet-152 − ResNet-101    | 15.9        | 0.6  | 0.038  |
> | HRNet [2]          | 256×192   | HRNet-W48 − HRNet-W32      | 35.08       | 0.7  | 0.020  |
> | ViTPose [3]        | 256×192   | ViT-Large − ViT-Base       | 219.65      | 1.4  | 0.006  |
> | PCT   [4]          | 256×192   | Swin-Large − Swin-Base     | 108.31      | 0.6  | 0.006  |
> | Simple Baseline [1] | 384×288   | ResNet-152 − ResNet-101    | 15.9        | 0.7  | 0.044  |
> | HRNet    [2]       | 384×288   | HRNet-W48 − HRNet-W32      | 35.08       | 0.5  | 0.014  |
> | Ours (RCNet)    |Work on all input size| Work on all backbone|**0.07**     |0.3-0.4| **4.28-5.71**|
>
> **Input scaling (same backbone params)**
>
> | Method | Backbone| Input Pair (Large − Small) | ΔGFLOPs | ΔmAP | mAP Gain / GFLOP  |
> |--------|---------|----------------------------|---------|------|-------------------|
> | Simple Baseline [1] | ResNet-101   | 384×288 − 256×192   | 15.5    | 2.2  | 0.142            |
> | Simple Baseline [1] | ResNet-152   | 384×288 − 256×192   | 19.6    | 2.3  | 0.117            |
> | HRNet  [2]         | HRNet-W32    | 384×288 − 256×192   | 8.9     | 1.4  | 0.157            |
> | HRNet  [2]         | HRNet-W48    | 384×288 − 256×192   | 18.3    | 1.2  | 0.066            |
> | Ours (RCNet)    | Work on all backbone| Work on all input size| **0.006** | 0.3-0.4|**50-66.7**|
>
> Moreover, we would like to re-emphasize our method advantages:
> - Consistant improvement on different HPE baselines including top-down, one-stage and bottom-up.
> - No retraining or architectural changes, acting as a plug-and-play post-hoc module, easy for deploymeny.
> - Enhanced calibration between predicted confidence and true pose quality (Tab.12 in Appendix H), crucial for safety-critical or human-centered applications.
>
> These advantages emerge from merely pose re-ranking, showing that ranking is a meaningful perspective but largely ignored in modern HPE. Ranking is so critical that even with perfect keypoints, modern HPE models still fall 15–18 mAP gap to perfection due to mis-ranking (see below table). This suggets a problem that inevitably needs to be addressed towards perfect pose estimation, and our method represents the first principled step in that direction.
>
>
> |Methods|HRNet-w32 |HRNet-w48 |ViTPose-Base |ViTPose-Large |EDPose|DEKR|
> |-------|----------|-------------|-----------|-------------|------|----|
> |mAP original| 74.4|75.1 | 75.8| 77.2| 71.6 | 67.1 |
> |mAP +gt keypoints| 82.1 |82.6 |82.7 | 83.2| 83.9|84.3 |
> |Gap to 100 mAP| 17.9| 17.4 | 17.3 | 16.8 | 16.1 | 15.7 |
> |mAR +gt keypoints | 97.5 | 97.5 | 97.5 | 95.0| 97.1 |94.6 |
>
>
> **[W2] Single-person vs. Multi-person**
>
> >**TL;DR** Our ranking still matters in single-person images, as it calibrates pose quality and confidence, a requirement for both single- and multi-person images. Since COCO mAP ranks predictions globally, better calibration still improves AP even for single-person cases.
>
> Even in single-person settings, ranking still matters. The underlying goal of our method is to calibrate confidence scores across poses to reflect true pose quality, independent of whether an image contains one person or many. COCO-style mAP is computed globally by taking all predicted poses from the entire dataset, both single-person and multi-person images, and sorting them by confidence before matching with ground-truth. Therefore, better calibration and ranking reduces the chance that low-quality poses appear above high-quality ones, improving AP regardless of whether images contain a single person or not.
>
> **[W3,W4] Citation and Caption Revision**
> We have revised and marked in blue color to the updated paper in the introduction, related works and the caption of Fig.1.

---

> > ### Author Response · Authors · 2025-11-21
> >
> > **[Q1] Other Assumptions?**
> >
> > >**TL;DR** A suitable distribution must be unimodal, mathematically tractable, and probabilistically meaningful. Laplacian is the simplest and most practical choice that satisfies all criteria.
> >
> > Beyond Laplacian and Gaussian, other distributional assumptions are in principle possible, but we found Laplacian to be the most suitable under our formulation.
> >
> > A suitable distributional assumption must satisfy several practical and theoretical constraints:
> >
> > - A unimodal distribution. The refined confidence should not have multiple possible peaks; otherwise, the refinement could assign several equally plausible confidence values to the same pose. A unimodal distribution guarantees a single, stable refinement direction.
> > - Analytic simplicity. To derive a usable loss function, the distribution must lead to a closed-form or easily computable objective. Our math formulation requires a closed-form evaluating $\int log p(x) dx$ or its variants, which becomes intractable or unstable if the distribution lacks analytic simplicity.
> > - Clear probabilistic meaning. Since confidence refinement aims to align predicted scores with true pose quality, the assumed distribution should have a clear probabilistic meaning (e.g., modeling absolute deviations or residual uncertainty).
> >
> > Following Occam’s razor, we deliberately avoid complicated distributions unless necessary. Laplacian and Gaussian families are natural choices because they are simple, interpretable, unimodal, and permit tractable closed-form losses.
> >
> > Other distributions, although theoretically possible, typically violate one of the above constraints. For example:
> >
> > **[Gamma distribution]**: more general than Gaussian/Laplacian, but lacks a closed-form integrand for the logarithmic terms required in training, making optimization impractical.
> >
> > **[Tukey–Lambda distribution]**: symmetric and interpretable through quantile functions but mathematically inconvenient; the quantile form complicates all derivations and prevents a clean closed-form loss.
> >
> > **[Gaussian Mixture Models]**: expressive but inherently multi-modal, producing multiple local maxima and violating the requirement of a single refinement direction.
> >
> > Given these considerations, the Laplacian assumption emerges as the simplest distribution that satisfies all constraints—unimodal, mathematically tractable, interpretable, and empirically effective in modeling pose-quality residuals.

---

### Official Review · Reviewer_VJvo · 2025-10-27

**Soundness:** 3
**Presentation:** 3
**Contribution:** 2
**Rating:** 4
**Confidence:** 5

**Summary:**

This paper argues that progress in 2D HPE has mostly focused on localization accuracy while overlooking confidence ranking, which directly affects mAP. The authors formalize pose confidence refinement as a pairwise ranking problem and derive a rank loss under Laplacian assumptions that upper-bounds the expected negative log-likelihood of correct orderings. They implement a lightweight self-attention module, RCNet, that refines pose confidence scores post-hoc without changing coordinates.

**Strengths:**

- Novel and well-motivated problem formulation: Shifting attention from keypoint localization to confidence ranking is a fresh perspective that addresses a real bottleneck in saturated benchmarks.
- Solid theoretical grounding: The derivation of the rank loss, with provable upper bound and convergence guarantee, is mathematically clear and well-explained.
- Lightweight, plug-and-play design.
- Empirical evidence: Gains are consistent across architectures and datasets.
- Clarity and writing quality: The paper is very readable, logically structured, and supported by informative figures.

**Weaknesses:**

- Incremental empirical improvement: the gains may not convince all readers that ranking alone constitutes a major advance.
- Assumptions could be restrictive: The Laplacian distribution may not generalize beyond the tested datasets; robustness to other settings is unclear.
- Broader significance: The framing is elegant but somewhat niche; most people care about 3D human pose/shape estimation instead of 2D. This project does not cover 3D pose or shape.

**Questions:**

- Is there a way to probabilistically justify the choice of Laplacian distribution?
- Where does the method stand against conformal-prediction-based baselines (e.g. CUPS by Zhang and Carlone).

CUPS: Improving Human Pose-Shape Estimators with Conformalized Deep Uncertainty

---

> ### Author Response · Authors · 2025-11-21
>
> **[W1] Incremental improvement.**
> >**TL;DR** Quantitative measurement on the improvement shows that our lightweight RCNet achieves far higher accuracy-per-cost efficiency than scaling input size or backbone parameters, while consistently improving a wide range of HPE baselines. This highlights mis-ranking as a key overlooked error source which alone accounts for the remaining 15–18 mAP gap even with perfect localization.
>
> We believe that determining whether the improvement is marginal is best guided by quantitative evidence rather than subjective impression. Thus, we quantitatively compare our method  with the two standard ways top-down HPE improves accuracy: scaling backbone size or input resolution, as Reviewer PMss suggested. These methods increase mAP only at the cost of large extra parameters or GFLOPs, whereas RCNet achieves +0.3–0.4 mAP with negligible computational overhead, achieving far higher mAP improvement per parameter and per FLOP than backbone or input scaling.
>
> **Backbone scaling (same input size)**
>
> | Method | Input  | Backbone Pair     | ΔParams (M) | ΔmAP | mAP Gain / M Params |
> |--------|--------|-------------------|-------------|------|----------------------|
> | Simple Baseline[1] | 256×192   | ResNet-152 − ResNet-101    | 15.9        | 0.6  | 0.038  |
> | HRNet [2]          | 256×192   | HRNet-W48 − HRNet-W32      | 35.08       | 0.7  | 0.020  |
> | ViTPose [3]        | 256×192   | ViT-Large − ViT-Base       | 219.65      | 1.4  | 0.006  |
> | PCT   [4]          | 256×192   | Swin-Large − Swin-Base     | 108.31      | 0.6  | 0.006  |
> | Simple Baseline [1] | 384×288   | ResNet-152 − ResNet-101    | 15.9        | 0.7  | 0.044  |
> | HRNet    [2]       | 384×288   | HRNet-W48 − HRNet-W32      | 35.08       | 0.5  | 0.014  |
> | Ours (RCNet)    |Work on all input size| Work on all backbone|**0.07**     |0.3-0.4| **4.28-5.71**|
>
> **Input scaling (same backbone params)**
>
> | Method | Backbone| Input Pair (Large − Small) | ΔGFLOPs | ΔmAP | mAP Gain / GFLOP  |
> |--------|---------|----------------------------|---------|------|-------------------|
> | Simple Baseline [1] | ResNet-101   | 384×288 − 256×192   | 15.5    | 2.2  | 0.142            |
> | Simple Baseline [1] | ResNet-152   | 384×288 − 256×192   | 19.6    | 2.3  | 0.117            |
> | HRNet  [2]         | HRNet-W32    | 384×288 − 256×192   | 8.9     | 1.4  | 0.157            |
> | HRNet  [2]         | HRNet-W48    | 384×288 − 256×192   | 18.3    | 1.2  | 0.066            |
> | Ours (RCNet)    | Work on all backbone| Work on all input size| **0.006** | 0.3-0.4|**50-66.7**|
>
> Moreover, we would like to re-emphasize below additional advantages of our method:
> - Consistant improvement on different HPE baselines including top-down, one-stage and bottom-up.
> - No retraining or architectural changes, acting as a plug-and-play post-hoc module, easy for deploymeny.
> - Enhanced calibration between predicted confidence and true pose quality (Tab.12 in Appendix H), crucial for safety-critical or human-centered applications.
>
> These advantages emerge from merely pose re-ranking, showing that ranking is a meaningful perspective but largely ignored in modern HPE. Ranking is so critical that even with perfect keypoints, modern HPE models still fall 15–18 mAP gap to perfection due to mis-ranking (see below table). This suggets a problem that inevitably needs to be addressed towards perfect pose estimation, and our method represents the first principled step in that direction.
>
> |Methods|HRNet-w32 |HRNet-w48 |ViTPose-Base |ViTPose-Large |EDPose|DEKR|
> |-------|----------|-------------|-----------|-------------|------|----|
> |mAP original| 74.4|75.1 | 75.8| 77.2| 71.6 | 67.1 |
> |mAP +gt keypoints| 82.1 |82.6 |82.7 | 83.2| 83.9|84.3 |
> |Gap to 100 mAP| 17.9| 17.4 | 17.3 | 16.8 | 16.1 | 15.7 |
> |mAR +gt keypoints | 97.5 | 97.5 | 97.5 | 95.0| 97.1 |94.6 |

---

> ### Author Response · Authors · 2025-11-21
>
> **[W2/Q1] Laplacian Assumptions and generalization.**
>
> >**TL;DR** Our Laplacian assumption is to provide a principled way to calibrate predicted confidences to true pose quality, turning our rank loss into an order-aware CRPS. Because the calibration design is not dataset-specific, our method also generalizes to CrowdPose and remains stable across different calibration metrics. In addition, the ranking perspective suggests possible extensions to other Learning-to-Rank tasks, such as information retrieval or recommendation.
>
>
> **(1) Probablistic Justification on Laplacian Assumption.**
>
> We give a full mathematical and probablistic analysis of Laplacian assumption on Supp.G. We briefly summarise the rationale below.
>
> Our choice of a Laplacian distribution is motivated by the empirical shape of pose-quality errors and by the probabilistic forecasting theory. We model the residual between the predicted pose score and its true quality (OKS target) as a symmetric, light-tailed error around zero. Under this assumption, a Laplacian/Gaussian distribution is a natural choice for the conditional distribution.
>
>
> We further find that Laplacian assumption stems from probabilistic-forecasting theory, where models assign calibrated confidence to predictions. Such systems rely on proper scoring rules that reward forecasts matching observed data. The most widely used rule is the Continuous Ranked Probability Score (CRPS), shown as follow
> $$
> CRPS(F,q)=\int (F(x) - I(x \ge q))^2dx,
> $$
> where $F$ is the predicted CDF, and  $I(x \ge q)$ is an emprical estimation of the groud truth CDF when one observed $q$. It contains a closed-form estimation as shown by the following equation,
> $$
> CRPS(F, q) = E_{F}[X-q] - \frac{1}{2} E_{F}[X-X'],
> $$
> where $X$ and $X'$ are i.i.d sampled in $F$.
>
> **Under a Laplacian assumption, our rank loss is equivalent to an order-aware CRPS, giving a principled probabilistic basis for refining the predicted pose scores toward their true quality.**  The full derivation and intuition are provided in Supp. G.
>
> Empirically, we show below table. Among all variants, only the Laplacian formulation consistently improves performance, making it the most effective choice.
>
> | Method        | Original | CRPS | Gaussian | Ours (Laplacian) |
> |---------------|------|------|----------|------------------|
> | HRNet [3]     | 74.4 | 69.5 | 74.6     | **74.7**         |
> | ViTPose [4]   | 75.8 | 69.8 | 75.9     | **76.0**         |
> | GroupPose [5] | 72.1 | 62.5 | 72.3     | **72.4**         |
> | EDPose [6]    | 71.6 | 61.9 | 71.9     | **72.0**         |
> | DEKR [7]      | 67.1 | 67.9 | 68.6     | **69.0**         |
> | PINet [8]     | 67.4 | 68.3 | 68.9     | **69.2**         |
>
>
>
>
> **(2) The Generalization of the Laplacian Assupmtion.**
>
> We'd like to argue our method including Laplacian Assumption is generalizable from 3 perspective.
>
> First, our method could be generalized to different dataset. The Laplacian assumption aims to calibrate confidence values with the dataset’s definition of pose quality.  This definition only relies on two quantities:(1).the confidence score predicted by the HPE model and (2).a measure of pose quality provided by the dataset (e.g., GT-based evaluation). This form of the definition is completely generic as any dataset that defines pose quality can be plugged into the same ranking formulation. Therefore, our approach applies broadly as long as a dataset provides a way to measure pose quality.
>
> Empirically, RCNet continues to improve all baselines on CrowdPose (Table 3 in the main paper), a dataset substantially different from COCO in both crowd density and annotation style.
>
>
> Second, a side benefit of the laplacian assumption is our method does not only work on ranking-based metrics. As discussed, it better aligns the confidence scores with the true pose quality, as shown in the table below, meaning it also improves calibration metrics.
>
>
> | Method           | Pearson Corr. ↑ | AUSE-OKS (%) ↓ |
> |------------------|------------------|-----------------|
> | HRNet | 0.414 | 1.42 |
> |  HRNet+Ours            | **0.547**       | **1.31**       |
> | EDPose | 0.430 | 1.98 |
> | EDPose+Ours            | **0.446**        | **1.87**      |
> | DEKR | 0.241 | 2.61 |
> | DEKR +Ours            | **0.326**      | **1.03**       |
>
>
>
> Lastly, by formulating our approach from a ranking-driven perspective, the methodology naturally extends beyond pose estimation and holds potential for a broad class of learning-to-rank problems. Such tasks aim to order items by relevance rather than predict labels or scores in isolation. Representative examples include information retrieval (e.g., Learning to Rank for Information Retrieval [9]), recommender systems (e.g., Learning to Rank for Recommender Systems [10]), and image retrieval (e.g., Image Retrieval Based on Learning to Rank and Multiple Loss [11]). This connection highlights the generality of our formulation and its applicability to other ranking-oriented domains.

---

> > ### Author Response · Authors · 2025-11-21
> >
> > **[W3] Broader Significance.**
> >
> > >**TL;DR** As an upstream task for 3D pose estimation, every improvement on 2D pose estimation will eventually turn to benefit the 3D pose community. Improving 2D pose confidence calibration benefits 3D lifting and 3D self-supervised training pipelines. Recent 3D metric start to rely on confidence-based mAP metrics, further increasing the relevance of our approach.
> >
> >
> > While our work focuses on 2D HPE, its impact naturally extends to 3D human pose and shape estimation.
> > **(1) It benefits lifting models in 3D pose.**
> >
> > Many 3D pipelines still follow a 2D→3D design [5], where a 3D lifter takes 2D keypoints (and often their confidences) from an off-the-shelf 2D pose estimator as input. Inaccurate or poorly ranked 2D poses therefore directly limit 3D accuracy [6], especially in realistic multi-person settings where selecting the correct 2D hypothesis per person is critical (e.g., outdoor datasets such as 3DPW).
> >
> > **(2) It benefits self-supervising methods in 3D Pose.**
> > Second, numerous weakly/self-supervised 3D approaches rely on 2D keypoints as supervision signals [7] or intermediate representations, so improvements in 2D pose quality and confidence calibration translate into more reliable and stable 3D training.
> >
> > **(3) It benfits mAP-style metric in 3D Pose.**
> >
> > Finally, 3D pose metric have begun adopting AP / mAP-style metrics in 3D space [8], which, just like COCO, depend on confidence ranking rather than localization alone. This suggests that confidence scores and their ranking are becoming increasingly important for 3D evaluation as well.
> >
> > Thus, although our method is instantiated on 2D HPE, a lightweight, architecture-agnostic module that improves confidence ranking is broadly relevant to current and emerging 3D pose and shape pipelines.
> >
> > **Q2. Comparison with CUPS and discussion on conformal prediction method.**
> >
> > >**TL;DR** CUPS relies on set-valued conformal prediction and adversarial score learning designed for 3D multi-hypothesis settings, which do not genearlly apply to 2D HPE; Design transferable to 2D HPE reduces to our CRPS baseline (see response to W2/Q1)
> >
> >
> > **(1). Fundamental problem mismatch (set-prediction 3D vs. single-prediction 2D).**
> >
> > CUPS is designed for 3D human mesh reconstruction, where the model inherently produces multiple stochastic 3D candidates for a single input. Conformal prediction is then used to produce a set-valued uncertainty region, and the size of this set signals confidence. In contrast, 2D HPE outputs exactly one pose with exactly one confidence score and there is no multi-hypothesis randomness to which conformal prediction can be applied. Thus, the core mechanism of CUPS related to conformal prediction, cannot be applied in 2D HPE.
> >
> > **(2). Their score-learning objective does not translate to 2D.**
> >
> > CUPS trains a score network using a GAN-like min–max objective jointly with the 3D pose estimator. This formulation is needed because their confidence score measures uncertainty over estimated pose smoothness, not only pose accuracy, so there is no simple ground-truth supervision available.
> >
> > However, this design is not applicable for 2D HPE, because in 2D HPE, confidence score correspond directly to OKS-based pose quality (accuracy). OKS provides a well-defined ground truth target for supervising confidence, and no adversarial training is required in 2D HPE.
> >
> > Our method instead provides a theoretically derived mapping aligning confidence distributions with OKS errors through a closed-form formulation, which is more suitable for 2D HPE methods and more training-friendly .
> >
> > **(3). A scalar analogue of CUPS already reduces to our CRPS baseline.**
> > If we strip away the set-valued aspect of CUPS and the GAN-like min–max training objective, CUPS reduces to calibrating the confidence to the pose quality (in our case, OKS), which is exactly what our CRPS baseline already implements. As shown in Table 10 (Appendix G) and also presented in the response to W2/Q1, this baseline does not improve mAP consistently, while our ranking head consistently does.
> >
> >
> > Reference:
> >
> > [1]Simple Baselines for Human Pose Estimation and Tracking, ECCV'18
> >
> > [2]Deep High-Resolution Representation Learning for Human Pose Estimation, CVPR'19
> >
> > [3]Simple Vision Transformer Baselines for Human Pose Estimation, ICCV/NIPS'22
> >
> > [4]Human Pose as Compositional Tokens, CVPR'23
> >
> > [5]MPL: Lifting 3D Human Pose from Multi-view 2D Poses, ECCV'24
> >
> > [6]Learning 2D Human Poses for Better 3D Lifting via Multi-Model 3D-Guidance, ACCV'24
> >
> > [7]Self-Supervised 3D Human Pose Estimation via Part Guided Novel Image Synthesis, CVPR'20
> >
> > [8]MV-SSM: Multi-View State Space Modeling for 3D Human Pose Estimation, CVPR'25
> >
> > [9]Learning to Rank for Information Retrieval. Foundations and Trends in Information Retrieval, 2009.
> >
> > [10]Learning to Rank for Recommender Systems. ACM RecSys, 2015.
> >
> > [11]Image Retrieval Based on Learning to Rank and Multiple Loss. ISPRS Int. J. Geo-Inf., 2019.

---

> ### Comment · Reviewer_VJvo · 2025-11-27
>
> I do not have further questions. I reassure the author that, after considering the rebuttal and discussion with other reviewers, I will update my score accordingly.

---

> > ### Author Response · Authors · 2025-11-28
> >
> > Thank you very much for your follow-up comment. We appreciate your careful reading of both the paper and the rebuttal, as well as your willingness to reconsider your assessment after discussing with the other reviewers. If there are any additional points that would benefit from clarification, we would be more than happy to provide further details. We sincerely thank you again for the time and thoughtful feedback.

---

### Official Review · Reviewer_7UCn · 2025-10-28

**Soundness:** 3
**Presentation:** 2
**Contribution:** 3
**Rating:** 6
**Confidence:** 3

**Summary:**

While 2D HPE models localize keypoints well, their confidence scores are poorly ranked, hurting mAP.  This work treats confidence ranking as a pairwise ordering problem and derive a rank loss that guarantees better ordering when minimized.  The 0.07-M-parameter RCNet refines these scores post-hoc without touching the keypoint coordinates.  Plugged into strong COCO and CrowdPose baselines, it boosts mAP by 0.3–1.8 points, showing that ranking alone is an overlooked yet effective lever for higher pose-estimation performance.

**Strengths:**

The author's perspective on thinking is quite novel. Both experimental and theoretical analysis are relatively sufficient.

**Weaknesses:**

1. The writing is not very easy to understand, and I am actually unclear about the underlying mechanism why better ranking can improve the performance of pose estimation algorithms.

2. From a technical perspective, there is a lack of specific descriptions on how to collaborate with related algorithms, as well as a lack of complexity analysis.

3.Because in some scenarios, the improvement is small and should have statistical significance analysis.

**Questions:**

Q1 “While existing HPE models predict confidence scores, they rarely optimize them for ranking, leading to performance that remains only modestly above random” This claim looks strange.

Q2 Why are the confidence scores of the two pose estimations not independent.

Q3 Why better ranking can improve the performance of pose estimation algorithms.

Q4 Why can RCNet exchange  sample order? It is obvious that the network only outputs scores.

---

> ### Author Response · Authors · 2025-11-21
>
> **[W1/Q3] Why Ranking Improves Pose Estimator Performance？**
>
> > **TL;DR:** Better confidence ranking matters because pose estimation depends on trustworthy confidence scores, reflected by COCO mAP metric. The metric matches higher-ranked poses with ground-truth first, so more accurate ordering directly yields higher true positives and better performance.
>
>
> **(1). Why Ranking Should Affect the HPE Performance?**
>
>
> Ranking matters because pose estimation depends not only on accurate keypoint localization but also on confidence scores that faithfully reflect the reliability of each predicted pose.
>
> These confidence scores directly influence downstream decisions, such as tracking, action recognition, and safety-critical systems, where knowing which poses to trust is essential.
>
> A pose estimator that localizes well but miscalibrates its confidence can still behave unpleasantly.
>
> Therefore, producing a confidence-based ordering that aligns with actual pose quality is an essential component of effective human-pose estimation.
>
> **(2). How Ranking Affect the HPE Performance?**
>
> Given this dual objective of localization and confidence estimation, the COCO evaluation metric mAP naturally incorporates both localization and ranking. Conceptually, mAP operates as follows:
>
> - Step 1, the pose estimator outputs a set of poses, each with a confidence score.
> - Step 2, all predictions are sorted by confidence (highest first).
> - Step 3, following this ranked list, each prediction is matched to the closest **remaining** ground-truth person (Greedy Matching).
> - Step 4, if the error (i.e., L2-distance between poses) is below a threshold, it counts as a correct detection (TP); otherwise, as a false alarm (FP).
>
> This process implies a critical mechanism. **Higher-confidence poses get first access to the limited ground-truth matches.** Therefore, if a slightly inaccurate pose is mistakenly ranked above a more accurate one, the better pose may lose its chance to match a ground truth—lowering mAP.
>
> Ranking therefore matters because matching is priority-based. Correctly ordering poses by their quality ensures that the most accurate poses are evaluated first and matched to ground truth, increasing true positives and reducing harmful mismatches.
>
> This is why improving ranking directly impacts performance.
>
>
>
> **[W2] Collaboration with Related Algorithms and Analysis**
>
> >**TL;DR** RCNet is a plug-in, post-hoc confidence refiner. It takes any HPE model’s predicted poses and confidence scores as input and outputs refined confidence values. It adds only a negligible computation cost, making integration with related algorithms straightforward and efficient.
>
> Our method is intentionally designed to collaborate with existing HPE algorithms in a minimal and principled way. RCNet works as a lightweight post-hoc module that operates on the outputs of any pose estimator (top-down, bottom-up, or one-stage). It requires only two inputs from the baseline model:
>
> - the predicted pose coordinates
> - the initial confidence score for each pose.
>
> Given these, RCNet produces refined confidence scores with better ranking. This means RCNet can be attached to any existing model without neither re-training nor affecting compatibility of the base HPE method. In practice, using RCNet is as simple as adding one extra forward pass on the set of predicted poses.
>
> We provided the detailed additional complexity table below. The RCNet is computationally negligible and easy to integrate into any algorithm.
>
> |Training Mem (MB) | Inference Mem (MB) | Throughout (pose per batch) | Latency (ms per pose) | GFLOPs | Model Size (M)|
> |------------------|--------------------|-----------------|------------|---------|-----|
> | 665 | 408 | 128| 2.25 x 10^-4| 0.006| 0.07 |
>
>
> **[W3] Statistical Analysis for Improvement**
> >**TL;DR** RCNet results are highly consistent across random seeds because COCO evaluation has very low variance and RCNet is fully deterministic. Thus, the consistent +0.3–0.4 mAP gains are systematic rather than due to chance.
>
> We conducted three runs with different random seeds and observed no variability in performance. This is expected for several reasons.
> - On COCO, mAP variance for a fixed training schedule and pretrained model is typically extremely small (often <0.1), so repeated runs rarely introduce meaningful differences.
> - RCNet operates purely as a post-hoc deterministic refinement during inference, without any stochastic optimization, so its output is stable across runs.
> - The +0.3 to +0.4 mAP improvement appears consistently across multiple top-down/one-stage backbones, making it very unlikely that the gains arise from random fluctuations.
>
> We will release full training/inference code and pretrained models upon acceptance. We ensure all results are fully reproducible.

---

> > ### Author Response · Authors · 2025-11-21
> >
> > **[Q1] Clarity of One Claim**
> > >**TL;DR** Existing HPE models output confidence scores, but these scores poorly reflect true pose quality, because they are not explicitly trained to produce a reliable rankinig of pose hypotheses.
> >
> > Existing HPE models do produce confidence scores, but these scores often fail to reflect the true quality of the predicted poses. Their confidence values are learned only as a by-product of detection or heatmap regression without any supervision that teaches the model how to order pose hypotheses according to actual pose quality.
> >
> > As a result, the predicted confidence does not reliably correlate with pose quality: the pairwise agreement between predicted confidence and true pose quality is typically just ~58–63%, barely above the 50% expected from random ranking (Tab. 1). The mentioned “performance” refers specifically to ranking accuracy, reflecting how well the predicted confidence ordering aligns with the ordering induced by pose-quality measurements.
> >
> > We agree that the original sentence was not written clearly, and this ambiguity may have caused confusion. The revised sentence has been updated at lines 202–204 and marked with blue color. We copy it below for your convenience.
> > ```
> > Existing HPE models produce confidence scores, but because these scores are not explicitly trained to reflect the correct ordering of pose quality, the resulting ranking accuracy is only slightly better than random when evaluated pairwise.
> > ```
> >
> > **[Q2] Why Are the Confidence Scores of Two Poses Dependent?**
> >
> > >**TL;DR** Pose confidences are dependent because (1) poses in the same image share contextual cues, (2) a single model must assign scores coherently, and (3) COCO’s matching/selection steps compare confidence across poses, requiring scores to be mutually consistent rather than independent.
> >
> > **(1) Poses within the same image are inherently correlated.**
> > Given the same image input, their representations and errors are statistically dependent, because interactions, overlaps, and contextual cues couple their representations. This dependence has even been explored by previous HPE method such as I²RNet [1].
> >
> > **(2) A single model must assign confidence scores in a coherent way.**
> > Confidence is not an arbitrary number per pose, as it is meant to reflect pose quality under a shared scoring function. If two poses have similar quality but receive inconsistent scores (e.g., one gets 0.1 and another gets 0.9), the values cannot be meaningful. Therefore, the scores must be jointly calibrated so that confidence scores are comparable across poses.
> >
> > **(3) The COCO evaluation protocol directly ties confidence to pose quality.**
> > In NMS, bottom-up grouping, and especially COCO’s greedy matching, multiple predicted poses directly compete for the same person. These procedures rely on comparing confidence values across poses, which means the scores must be mutually coherent rather than treated independently.
> >
> > Therefore, in practice, pose confidences are inherently dependent.
> >
> >
> >
> > **[Q4] Why Score Changes Sample Ordering**
> >
> > >**TL;DR** “Sample order” refers to the confidence-based priority for matching predictions to ground truth in COCO evaluation. Since this priority is determined entirely by confidence scores, refining the scores naturally changes the matching order.
> >
> >
> > We believe the misunderstanding comes from what “sample order” refers to in  COCO evaluation. It refers to the priority order in which predictions are allowed to claim ground-truth persons during mAP computation.
> >
> > This priority is entirely determined by the confidence scores: higher-confidence predictions get to match first, and lower ones receive whichever ground-truth instances remain. Therefore, adjusting the confidence scores is exactly equivalent to adjusting this selection order. If more True Positives (TPs) are ranked above False Positives (FPs), the resulting mAP is higher.
> >
> > This is exactly why adjusting the confidence scores matters, which aligns the ordering with the actual pose quality, thus enhancing COCO mAP. Beyond performance, producing confidence values that are more meaningful and reliable benefits human-centered downstream applications.
> >
> > Thus, a post-hoc confidence refinement module like RCNet can change the sample ordering in COCO evaluation, without altering poses.
> >
> >
> > Reference:
> >
> > [1] I2R-Net: Intra- and Inter-Human Relation Network for Multi-Person Pose Estimation, IJCAI'22

---

### Official Review · Reviewer_PMss · 2025-11-03

**Soundness:** 3
**Presentation:** 3
**Contribution:** 2
**Rating:** 2
**Confidence:** 5

**Summary:**

- The paper focuses on the problem of multi-person 2D pose ranking.
- The method trains a lightweight ranking network (0.07M parameters) to re-rank a baseline model’s multi-person pose predictions, using OKS-derived surrogate targets.
- The evaluations are done across multiple datasets: COCO, Crowdpose and cover both top-down and bottom-up baselines.
- RCNet consistently improves mAP across baselines and yields better person-ranking across scenarios.

**Strengths:**

- The paper is well written, organized and easy to follow. The key ideas are presented upfront, and the technical details are clear.
- The focus on ranking pose predictions is novel; to the best of my knowledge, no prior learning-based solution directly addresses this task.
- The evaluation spans multiple datasets and conditions (general and crowded) and reports RCNet’s improvements across diverse baselines.
- RCNet’s formulation is supported by strong theoretical explanations and intuitions. I really appreciate the rigorous proofs for the propositions and theorems in the appendix.

**Weaknesses:**

- Marginal Improvements: The method adds only +0.3 mAP on COCO val with ViTPose-Base at $256\times192$—likely within run-to-run variance; longer training of the base checkpoint could close the gap. Moreover, gains are reported only on lighter backbones; larger top-down models with greedy ground-truth–to–prediction matching typically have sharper pose ranking, so improvements on heavier backbones are likely even smaller. Please consider reporting improvements on the larger variants of ViTPose at 384 resolution.

- Ranking gains don’t change recall: By design, RCNet reorders scores and leaves detections/coords unchanged; mAR is unaffected, limiting impact when localization is already the bottleneck (in crowding, occlusions or truncations).

- “Near-random” baseline ranking claim: Sec. 4 explains the near-random baseline claim for ranking. The paper shows ~58–63% pairwise accuracy for several methods, but this hinges on pair sampling and the matching protocol. Please correct me if I am wrong under the top-down COCO greedy-matching protocol, ranking should not be near-random (due to predicted confidence based sorting) and should not materially affect mAP. Please clarify the setup.

- Limited qualitative results: Fig. 4 in the main paper is the only qualitative example and does not show how precision improves over the baseline. Please revise to clearly visualize the benefit from correct re-ranking (e.g., highlight corrected orderings and the resulting AP change for the shown image). I am especially interested in the gains of RCNet over confidence-based sorting (a reasonable proxy for OKS); since RCNet is supervised with OKS-derived targets, confidence sorting may achieve similar effects—please disentangle and quantify this gap.

**Questions:**

My questions are primarily centered on the weaknesses mentioned above.

1. Please evaluate on RCNet with heavier backbones and report the gains.
2. The core contribution only appears to improve mAP and therefore has limited benefits. Please consider sharing your perspective on this.
3. Please expand on the random ranking claim.
4. Improved qualitative results.

---

> ### Author Response · Authors · 2025-11-21
>
> **[W1/Q1]Question on Improvement Significance**
>
>
> > **TL;DR:**  Our method consistently provides meaningful gains with negligible overhead in required experiments, far beyond run-to-run variance confirmed by different random seeds and longer training time. In contrast, simply extending training epochs, scaling model size or input resolution yields far lower efficiency and does not resolve persistent ranking errors. Our method tackles this orthogonal, long-overlooked ranking problem that scaling cannot fix, making its improvement not trivial.
>
> **(1). Additional experiments on heavier backbones and different input sizes**
> Following the reviewer’s suggestion, we report the performance of several representative top-down models under (1) an additional 10-epoch continual training schedule with heavier backbones and (2) our RCNet module. ViTPose does not provide official 384×288 pretrained checkpoints, so we follow the official release (256×192) for the larger ViT backbones, and we additionally include HRNet models which do provide 384×288 pretrained weights.
>
> From results shown below, we observe two consistent phenomena:
> - Continual training (even 10 extra epochs) does not improve performance and often overfits.
> - RCNet consistently improves strong top-down models by +0.3–0.4 mAP.
>
> Across all tested backbones and input sizes, the ranking mismatch persists even with stronger localization (Table 1, 5), and RCNet consistently delivers +0.3–0.4 mAP improvements. We conducted three runs with different random seeds and observed no variability in performance.
>
>
> | Model         | Input Size  |# of Params | GFLOPs |Baseline AP | +10 epoch training | +Ours  |
> |---------------|-------------|------------|--------|------------|--------------------|--------|
> | HRNet-W32     | 256×192     | 28.59      | 7.1    | 74.4       |  74.3  (-0.1)  | 74.8 (+0.4)|
> | HRNet-W32     | 384×288     | 28.59      | 16.0   | 75.8       |  75.6 (-0.2)   | 76.1 (+0.3)|
> | HRNet-W48     | 256×192     | 63.68      | 14.6   | 75.1       |  75.0 (-0.1)   | 75.5 (+0.4)|
> | HRNet-W48     | 384×288     | 63.68      | 32.9   | 76.3       |  76.0 (-0.3)   | 76.7 (+0.4)|
> | ViTPose-Base  | 256×192     | 89.99      | 16.43  | 75.8       |  74.6 (-0.8)   | 76.1 (+0.3)|
> | ViTPose-Large | 256×192     | 308.55     | 58.17  | 77.2       |  77.0 (-0.2)   | 77.5 (+0.3)|
> | ViTPose-Huge  | 256×192     | 637.21     | 121.04 | 77.9       |  77.7 (-0.2)   | 78.2 (+0.3)|
>
>
> **(2). Quantifying the RCNet's Improvement**
>
> To assess improvement significance, we rely on quantitative comparisons instead of subjective intuition. We present tables that directly compare mAP gains against the added parameters and FLOPs across different backbone sizes and input resolutions.
>
> Based on these comparisons, RCNet delivers a stable +0.3–0.4 mAP gain at negligible cost, outperforming backbone or input scaling in efficiency. Our method offers substantially higher mAP-per-parameter and mAP-per-FLOP efficiency than increasing backbone size or input resolution. Its gains hold across diverse architectures and input settings, indicating that model scaling cannot resolve the underlying confidence-ranking issue. This underscores that ranking is a separate, long-overlooked dimension of HPE.
>
> **Backbone scaling (same input size)**
>
> | Method | Input  | Backbone Pair     | ΔParams (M) | ΔmAP | mAP Gain / M Params |
> |--------|--------|-------------------|-------------|------|----------------------|
> | Simple Baseline | 256×192   | ResNet-152 − ResNet-101    | 15.9        | 0.6  | 0.038  |
> | HRNet           | 256×192   | HRNet-W48 − HRNet-W32      | 35.08       | 0.7  | 0.020  |
> | ViTPose         | 256×192   | ViT-Large − ViT-Base       | 219.65      | 1.4  | 0.006  |
> | PCT             | 256×192   | Swin-Large − Swin-Base     | 108.31      | 0.6  | 0.006  |
> | Simple Baseline | 384×288   | ResNet-152 − ResNet-101    | 15.9        | 0.7  | 0.044  |
> | HRNet           | 384×288   | HRNet-W48 − HRNet-W32      | 35.08       | 0.5  | 0.014  |
> | Ours (RCNet)    |Work on all input size| Work on all backbone|0.07     |0.3-0.4| **4.28-5.71**|
>
> **Input scaling (same backbone params)**
>
> | Method | Backbone| Input Pair (Large − Small) | ΔGFLOPs | ΔmAP | mAP Gain / GFLOP  |
> |--------|---------|----------------------------|---------|------|-------------------|
> | Simple Baseline | ResNet-101   | 384×288 − 256×192   | 15.5    | 2.2  | 0.142            |
> | Simple Baseline | ResNet-152   | 384×288 − 256×192   | 19.6    | 2.3  | 0.117            |
> | HRNet           | HRNet-W32    | 384×288 − 256×192   | 8.9     | 1.4  | 0.157            |
> | HRNet           | HRNet-W48    | 384×288 − 256×192   | 18.3    | 1.2  | 0.066            |
> | Ours (RCNet)    | Work on all backbone| Work on all input size| 0.006 | 0.3-0.4|**50-66.7**|

---

> > ### Comment · Reviewer_PMss · 2025-11-27
> >
> > Thank you for the rebuttal. I appreciate the comparison with continued learning, that helps put RCNet gains in perspective.
> >  I am assuming the evaluations are consistent with Table 2, on COCO val set.
> >
> > Question: why don't the ViTPose (huge, large) baseline numbers match with the reported results from ViTPose paper (https://arxiv.org/pdf/2204.12484), Table 9? The reported numbers with RCNet are lower than the published baseline results.

---

> > > ### Author Response · Authors · 2025-11-28
> > >
> > > Thank you for the follow-up and for carefully checking the ViTPose results. We believe the discrepancy mainly stems from version mismatches in the MMPose/MMCV framework. ViTPose was developed on early MMPose 0.x commits, while the public repository does not specify the exact versions used for the reported numbers. As noted by the MMPose community, the transition from 0.x to 1.x introduced non–backward-compatible changes that can shift COCO AP and make older checkpoints unreproducible under newer versions (e.g., https://github.com/open-mmlab/mmpose/issues/1687).
> > >
> > > Because our method requires continual training and is plug-and-play, it is essential that all ViTPose variants are evaluated under the same detector, data split, framework version, and inference protocol. Therefore, all ViTPose results in our paper are obtained using the MMPose version that correctly reproduces the ViTPose-Base result.
> > >
> > > Under this unified setting, the conclusions remain unchanged: continual training does not improve ViTPose, whereas RCNet consistently yields +0.3–0.4 mAP at negligible cost. If anonymity permits, we are happy to share our evaluation setup and COCO-format JSON predictions to ensure full transparency.

---

> ### Author Response · Authors · 2025-11-21
>
> **[W2/Q2] Reordering Alone Can’t Fix Localization**
>
> > **TL;DR:** RCNet targets persistent ranking gap that remains even with ideal localization, as ranking and localization are complementary and equally important in HPE.    Although RCNet leave mAR unchanged, mAR reflects only localization, while mAP is more comprehensive and meaningful.
>
>
> Localization and confidence ranking are complementary dimensions of the HPE problem, and scaling the model cannot fix the ranking issue.Even when we provide perfect localization using ground-truth keypoints, mAR rises to 94–97% but mAP still stays 15–18 points below 100% across all evaluated models. This shows that a large portion of the remaining gap comes from confidence ordering rather than localization. Thus, improving ranking directly addresses the part of the problem that localization alone cannot close, highlighting why this long-overlooked dimension deserves dedicated methods such as ours.
>
> Overall, HPE involves three challenges, localization, detection, and confidence ranking, and mAR reflects only the first two. In contrast, mAP captures all three, making it a more comprehensive and demanding metric. Therefore, even when mAR does not change, improving mAP directly indicates better overall pose estimation quality and remains the key indicator of meaningful progress in HPE.
>
>
> |Methods|HRNet-w32 |HRNet-w48 |ViTPose-Base |ViTPose-Large |EDPose|DEKR|
> |-------|----------|-------------|-----------|-------------|------|----|
> |mAP original| 74.4|75.1 | 75.8| 77.2| 71.6 | 67.1 |
> |mAP +gt keypoints| 82.1 |82.6 |82.7 | 83.2| 83.9|84.3 |
> |Gap to 100 mAP| 17.9| 17.4 | 17.3 | 16.8 | 16.1 | 15.7 |
> |mAR +gt keypoints | 97.5 | 97.5 | 97.5 | 95.0| 97.1 |94.6 |
>
>
>
>
> **[W3/Q3] “Near-random” baseline ranking claim**
>
> > **TL;DR:** COCO evaluation sorts poses by confidence but never ensures that confidence reflects OKS. Greedy matching only uses this sorted order without revision. When we directly compare confidence vs. OKS on all pose pairs, SOTA HPE models still achieve only 58–63% correct ordering (random = 50%), confirming their weak fine-grained ranking.
>
> The key clarification is that COCO never enforces confidence scores to reflect OKS quality, even though it sorts predictions by confidence before greedy matching. We provide COCO evaluation protocol for clarity, which is used by the evaluation of all pose estimation methods regardless of top-down, bottom-up, or one-stage pipelines.
>
> **COCO evaluation protocol**:
> - Step 1. A pose estimator outputs a set of poses, each with a single confidence score.
> - Step 2. All poses are sorted by confidence (highest first).
> - Step 3. Iterating through this sorted list, each prediction is matched to the best unmatched GT by OKS **(Greedy Matching)**.
> - Step 4.If the best OKS exceeds the threshold, it is a TP; otherwise, an FP.
>
> Because COCO simply uses the model’s confidence outputs, the sorted order in Step 2 is not guaranteed to align with the true OKS ordering, as there is no such alignment supervision in HPE training. This alignment is exactly what we measure as ranking accuracy. Greedy matching (Step 3) only consumes this order when assigning TPs and FPs and it does not improve the confidence ranking. Therefore, it is likely that this alignment is only marginally better than random (statistically independent) ranking, even though the model still produces reasonable, though far from perfect performance. Our analysis therefore measures how well the confidence ordering aligns with OKS by exhaustively checking all pose pairs after matching. We confirm that, under this fine-grained definition, SOTA top-down models with ~75–76 mAP still achieve only 58–63% correct ordering (random = 50%). Thus, achieving stronger alignment through proper ranking is essentially the last missing piece towards pefect pose estimation.
>
>
> **[W4/Q4] Revise the visualization**
>
> We have revised Fig. 4 in the updated paper. In particular, we now focus on only one example but illustrates in detail how AP improves for this case.
> In short, after refining the ranking, true-positive (marked in green box) prediction obtains higher confidence score than false-positives (marked in red box), leading to enhanced AP performance.
> Another example is provided in the updated Appendix I. We are happy to add more examples if the reviewer finds them helpful.
>
> Reference:
>
> [1]Simple Baselines for Human Pose Estimation and Tracking, ECCV'18
>
> [2]Deep High-Resolution Representation Learning for Human Pose Estimation, CVPR'19
>
> [3]Simple Vision Transformer Baselines for Human Pose Estimation, ICCV/NIPS'22
>
> [4]Human Pose as Compositional Tokens, CVPR'23

---

### Author Response · Authors · 2025-11-21

Dear Reviewers,

We appreciate the reviewers’ recognition of both the conceptual novelty and practical impact of our work. Reviewers highlighted the paper’s
- Clarity and strong organization (PMss, Vjvo),
- Novelty and importance of addressing pose-confidence ranking (PMss, 7UCn, Vjvo, TScq),
- Solid theoretical foundations supported by rigorous analysis (PMss, 7UCn, Vjvo, TScq).
- Comprehensive experiments and consistent gains across datasets and baselines (PMss, 7UCn, Vjvo, TScq),
- lightweight, plug-and-play nature of RCNet (Vjvo, TScq).

We have uploaded a detailed response for each reviewer. To respect your valuable time, each answer begins with a concise TL;DR. The revised manuscript has also been updated, with new changes highlighted in blue.

Thank you very much for your time and effort. We look forward to a productive and constructive discussion in the rest of the discussion phase.

Sincerely,
The Authors

---

### Author Response · Authors · 2025-11-30
**Summary of Rebuttal and Public Discussion Feedback**

Dear Area Chair,

Thank you for taking over our paper under the unusual circumstances. Since reviewer scores have reverted to their pre-discussion values, we would like to briefly summarize how the rebuttal and subsequent public discussion evolved. This summary is intended both to help streamline your assessment and to highlight that **the major concerns raised by the initially more critical reviewers (PMss and VJvo) were effectively addressed during the discussion phase, according to their own publicly posted comments.**

**0. Summary of the Paper**

**[One-sentence summary]** This submission identifies confidence ranking as a missing yet essential component of human pose estimation and proposes a principled solution to address this overlooked problem.

**[Summary of our contribution]**
- Novel problem with a principled solution.
- Supported by rigorous theoretical analysis (boundedness and convergence).
- RCNet is a 0.07M-parameter, 0.006-GFLOPs, plug-and-play module that consistently improves performance across all HPE paradigms (top-down, bottom-up, one-stage, and both lighter/heavier backbone variants.)


**1.Reviewer Consensus on Strengths**

- **Novel Problem Formulation (All reviewers)**
All reviewers agree the paper identifies a genuinely new and overlooked 2D HPE.
- **Theoretical Soundness (PMss, VJvo, 7UCn)**
Reviewers highlight the method’s solid theoretical grounding, with clear derivations and rigorous proofs.
- **Empirical Strength with Consistent Gains (All reviewers)**
All reviewers observe consistent, repeatable improvements across datasets and diverse HPE baselines
- **Practical Utility: Lightweight, Efficient, and Compatible (VJvo, TScq, PMss)**
Reviewers emphasize RCNet’s negligible cost, plug-and-play design, and broad compatibility with existing models.
- **Clarity and Readability (PMss, VJvo, TScq)**
Reviewers describe the paper as clear, well written, and easy to follow.

**2. Resolution of Key Reviewer Concerns**

- **Magnitude and Significance of Improvements [PMss W1/Q1, VJvo W1, TScq W1]**
We added extensive comparisons demonstrating that RCNet achieves 50–66× higher mAP-per-GFLOP and 4–6× higher mAP-per-parameter efficiency compared to scaling backbones or input resolution. Even extremely large models benefit (+0.3–0.4 mAP).
- **Why Ranking Directly Affects Performance (mAP)? [7UCn W1/Q2]**
We clarified that COCO mAP is inherently order-dependent due to greedy matching. Even with perfect localization, modern HPE models still exhibit a 15–18 mAP gap caused solely by mis-ordering, confirming ranking as a core error mode.
- **Generalizable Laplacian Assumption [VJvo W2/Q1]**
We expanded Supp. G to show that under Laplacian assumptions, the proposed loss reduces to an order-aware CRPS, providing a principled probabilistic forecasting basis for aligning confidence with OKS. Because CRPS relies only on predicted scores and dataset-defined quality signals, the formulation naturally generalizes across datasets. Empirically, only the Laplacian-based version consistently improves mAP across models and datasets.
- **Other Probabilistc Assumptions [TScq Q1]**
We clarified a suitable distribution must be unimodal, mathematically tractable, and probabilistically meaningful. Laplacian is the simplest and most practical choice that satisfies all criteria.

**3.Reviewers Trajectory**


During the public discussion phase, both reviewers who initially raised the strongest concerns (PMss, VJvo) posted follow-up comments that indicate their earlier issues were effectively addressed by the rebuttal.

- Reviewer VJvo (initial score 4) indicated that they had no further questions and publicly stated that they would update their score after considering the rebuttal and discussion. This shows that the rebuttal fully resolved their earlier doubts. We quote the reviewer’s words below.
```
I do not have further questions. I reassure the author that, after considering the rebuttal and discussion with other reviewers, I will update my score accordingly.
```
- Reviewer PMss (initial score 2) expressed appreciation for the additional experiments and noted that they clarified the magnitude and context of RCNet’s improvements. Their only remaining follow-up concerned the ViTPose baselines, for which we provided a concrete explanation based on the evaluation protocol and MMPose 0.x -> 1.x differences. No new objections were raised after that exchange, before the discussion was halted. We quote the reviewer’s words below.
```
I appreciate the comparison with continued learning, that helps put RCNet gains in perspective.
```

We hope that the abrupt termination of the discussion phase, especially the ongoing exchange with PMss, will not disadvantage our submission. Thank you for considering this context together with the original reviews.

---

### Note · Program_Chairs · 2026-01-17
**Submission Desk Rejected by Program Chairs**

The following references in this submission do not refer to real documents and/or have major errors in bibliographic information:

 Aritra Ghosh, Mojtaba Kadkhodaie, Ateet Bhardwaj, Shashank Shekhar, and Stefano Ermon. Adafocal: Calibration-aware adaptive focal loss. In Advances in Neural Information Processing Systems 36 (NeurIPS), 2022.